# ON LEARNING REPRESENTATIONS FOR TABULAR DATA DISTILLATION

## ABSTRACT

Dataset distillation generates a small set of information-rich instances from a large dataset, resulting in reduced storage requirements, privacy or copyright risks, and computational costs for downstream modeling, though much of the research has focused on the image data modality. We study tabular data distillation, which brings in novel challenges such as the inherent feature heterogeneity and the common use of non-differentiable learning models (such as decision tree ensembles and nearest-neighbor predictors). To mitigate these challenges, we present **TDColER**, a tabular data distillation framework via column embeddings-based representation learning. To evaluate this framework, we also present a tabular data distillation benchmark, TDBench. Based on an elaborate evaluation on TDBench, resulting in 226,200 distilled datasets and 541,980 models trained on them, we demonstrate that **TDColER** is able to boost the distilled data quality of off-the-shelf distillation schemes by 0.5-143% across 7 different tabular learning models.

## 1 INTRODUCTION

Dataset distillation or dataset condensation is the process of creating a small set of extremely informative samples (usually synthetic) from a large dataset such that a model trained on this set will have predictive performance comparable to that of a model trained on the original large dataset (Wang et al., 2020; Yu et al., 2023). First, data distillation reduces data storage costs and can mitigate the privacy and copyright concerns involved in keeping around (and continuously utilizing) large amounts of raw data. Furthermore, the reduction in the data size implies a lower computational cost of model training, especially when multiple models need to be trained on any given dataset. The above advantages of dataset distillation also facilitate various applications. Continual learning, where we need to learn new tasks while avoiding forgetting older tasks sequentially, often makes use of a "replay buffer" of old task data to be used while learning new tasks to mitigate forgetting of the older tasks (Rolnick et al., 2019). Dataset distillation reduces the memory overhead of this replay buffer, allowing learning of a larger number of tasks without forgetting (Tiwari et al., 2022; Rosasco et al., 2022).

In federated learning, we need to train a model using data spread across multiple clients without ever moving the data between clients and reducing the communication overhead. Dataset distillation generates compact yet private synthetic data from the client data that can be safely exchanged for communication-efficient model training (Song et al., 2023; Goetz & Tewari, 2020; Zhou et al., 2020).

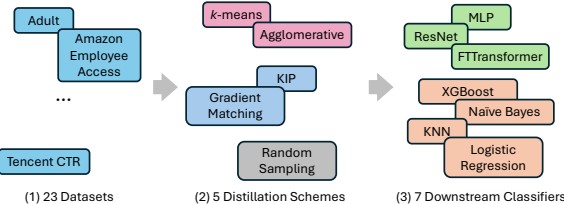

(1) 23 Datasets     (2) 5 Distillation Schemes     (3) 7 Downstream Classifiers

Figure 1: **Overview of** TDBench**.** The benchmarking suite allows for flexible choice of datasets, distillation schemes, and downstream models that enables for modular evaluation of any new distillation method.

While dataset distillation has been widely studied for image datasets (Cui et al., 2022; Yu et al., 2023), the equally important application to other data modalities is limited. The problem of tabular data distillation has received very little attention, though many real-world learning problems and applications involve tabular data (Guo et al., 2017; Clements et al.; Borisov et al., 2024). Various image data distillation schemes have been proposed in the literature, but their application to tabular data is not straightforward. First, all image data distillation schemes rely on the choice of a *differentiable* "backbone model." While differentiable

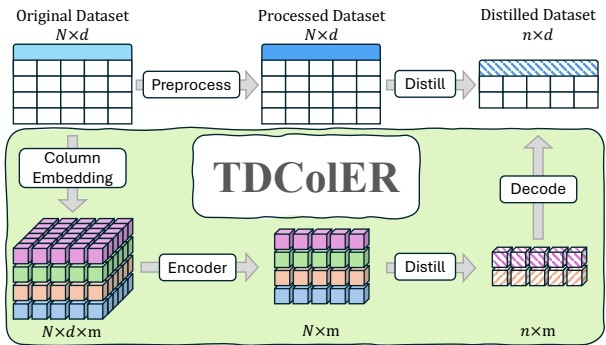

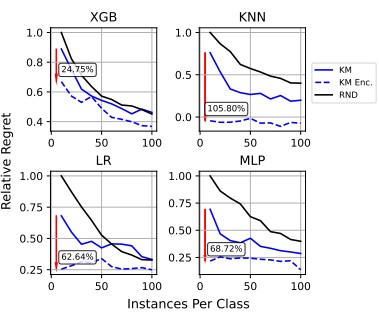

(a) **Overview of TDColER.** The top describes a *vanilla* distillation scheme that only uses standard preprocessing techniques before distillation. The highlighted box describes the proposed **TDColER**, which uses column embeddings after such preprocessing and encoder-decoder architectures to generate rich compact representations.

(b) Snapshot of downstream classifiers' performance increase when trained on data distilled $k$-means with and without **TDColER**. Throughout our experiments, we observe a performance increase from 0.5% to as large as 143%.

Figure 2: Proposed approach – **TDColER**: **T**abular **D**istillation Via **Col**umn **E**mbeddings based **R**epresentation Learning

neural network-based schemes are standard for images, a wide variety of non-differentiable models are used with tabular data, such as decision tree ensembles, nearest-neighbor models, and kernel machines. Second, almost all data distillation methods for images generate distilled data in the original pixel space. While pixels are homogeneous raw features of an image, the features in tabular data can be extremely heterogeneous, creating a mismatch between what the image data distillation methods are designed for and what we have as an inherent property of tabular data. Finally, it is standard to use vision-specific data augmentation schemes (such as rotation, reflection, cropping, and translation) to train the model on the distilled image data. Such standard augmentations are not available for tabular data, thus creating another discrepancy in the expected conditions for the problem.

**Our contribution.** In this paper, we study tabular dataset distillation and present a novel scheme to enhance the distilled data quality of multiple off-the-shelf data distillation schemes across various datasets, models, and distillation sizes. Specifically, we make the following contributions:

- We propose **T**abular **D**istillation via **Col**umn **E**mbeddings based **R**epresentation Learning or **TDColER** that can utilize modern neural-network architectures such as Transformers and graph neural networks to generate rich compact representations. **TDColER** improves the quality of distilled data compared to existing distillation schemes. Figure 2a provides an overview of our proposed **TDColER**.
- We present TDBench, a **T**abular **D**istillation **Bench**mark with 23 tabular datasets, 7 model classes, and 4 distillation schemes. We present an overview of TDBench, an extensible and modular framework for measuring various aspects of data distillation on tabular data, in fig. 1.
- With the elaborate evaluation of our proposed **TDColER** on TDBench, resulting in over 226,200 distilled datasets and 541,980 model trainings, we show that, on aggregate across all datasets, **TDColER** improves upon direct application of off-the-shelf distillation method on tabular data by 0.5-143% in terms of the distilled data quality across all models at the smallest distillation of 10 instances-per-class. Figure 2b presents a snapshot of our results.
- Based on our thorough evaluation, we present various insights regarding tabular dataset distillation, such as (i) $k$-means clustering in the learned representations make for an extremely favorable distillation scheme, (ii) transformer-based tabular data representations obtain the highest distilled data quality on aggregate, while (iii) graph neural network based tabular data representations perform slightly worse than transformers but are significantly more parameter efficient.

## 1.1 RELATED WORK

Dataset distillation was introduced by Wang et al. (2020) as a bilevel optimization problem (Feng et al., 2024) and has been widely studied in the context of image data distillation. Most methods can

be categorized into approaches that match the original data by (i) backbone model performance, (ii) backbone model parameters, or (iii) backbone representation distributions (Yu et al., 2023). Wang et al. (2020) minimized performance differences between the original and distilled data, while Nguyen et al. (2021) introduced kernel-induced points (KIP) using kernel ridge regression with a neural tangent kernel (Jacot et al., 2018). Alternatively, methods have focused on parameter or gradient matching (Zhao et al., 2021; Lee et al., 2022; Jiang et al., 2023; Cazenavette et al., 2022). Gradient matching (Zhao et al., 2021) aligns model gradients between original and synthetic data, while trajectory matching (Cazenavette et al., 2022) minimizes discrepancies between entire training trajectories. Other approaches include distribution matching (Zhao & Bilen, 2023), which aligns per-class means, and cross-layer feature embedding matching (Wang et al., 2022). However, the abovementioned methods rely on differentiable backbones, limiting cross-architecture generalization (Cui et al., 2022; Nguyen et al., 2021). As a result, research has focused primarily on images, leaving *tabular data distillation* largely unexplored (Medvedev & Dyakonov, 2021). We address this gap by proposing a more general distillation framework.

Dataset distillation aligns with coreset selection (Feldman, 2020), which aims to reduce data size, typically selecting real data instances (potentially risking privacy). In contrast, distillation generates synthetic data beyond the real data manifold. Notably, coreset selection is a subset of dataset distillation, where the synthetic data lies on the real data manifold. Generative modeling (Goodfellow et al., 2020; Kingma & Welling, 2013) is another related area, usually focused on generating realistic data. In dataset distillation, the goal is to generate informative rather than realistic samples. Recently, Cazenavette et al. (2023) demonstrated how generative modeling can be used to seed the dataset distillation process, arguing that distillation methods should be applied to a latent representation instead of the pixel space. This is aligned with our proposal, in which we demonstrate that distillation in the latent space is critical to obtaining meaningful distilled data quality with tabular datasets. However, the proposed Generative Latent Distillation(GLaD) scheme is very focused on generative vision models, requiring a careful choice of the latent representation from within the model for trade-off in *realistic* distilled data or *expressivity*, thus limiting cross-architecture generalization.

Cui et al. (2022) benchmarked several distillation methods and found trajectory matching (Cazenavette et al., 2022) to be most effective, followed by KIP (Nguyen et al., 2021). Coreset methods, like $k$-means clustering, also outperformed many model-based distillation techniques, which we corroborate. We focus on GM and KIP due to the high computational overhead of trajectory matching and omit data augmentation due to its limited applicability to tabular data. As noted before, data augmentation is not standard with tabular data, and we do not consider it in our evaluation with TDBench.

## 2 TABLE DISTILLATION

Data distillation has been primarily studied in the context of images where each data point is composed of a homogeneous set of features – pixels – and the downstream models are neural networks. The two main distinctions with tabular data distillation are: (i) **Feature Heterogeneity**: Features in tabular data are usually heterogeneous and can have vastly different meanings, making it challenging to generate appropriate feature aggregations as usually done with neural networks. This is further exacerbated by the common presence of missing values. (ii) **Model Agnosticity**: For tabular data, the downstream model that will use the distilled data can be quite varied, with decision-tree-based models often being quite successful (Grinsztajn et al., 2022), while linear and nearest-neighbor models are used for interpretability.

---
**Algorithm 1:** Distill original data $S$ with $N$ samples given a *preprocessor* $P : \mathbb{R}^r \times \mathbb{C}^c \to \mathbb{R}^D$ and a *distiller* $F : \mathbb{R}^{N \times D} \times Y^N \to \mathbb{R}^{n \times D} \times Y^n$.

---
**1** $\tilde{S} \leftarrow \{(P(x), y) \, \forall (x, y) \in S\}$   // Preprocess
**2** $R \leftarrow F(\tilde{S})$               // Distill
**3 return** $R$

---

Various increasing competitive neural-network-based models have also been developed for tabular data (Borisov et al., 2024; Gorishniy et al., 2021; McElfresh et al., 2023; Grinsztajn et al., 2022). However, in the most common cases, we cannot assume that the downstream model is differentiable and thus will be unable to perform a downstream model-specific distillation via the common bilevel formulation of the problem. The distillation has to be model-agnostic, which means that we have to retain as much of the information in the original data as possible since we do not know a priori what information the downstream model might leverage.

We will consider a classification dataset $S = \{(x_1, y_1), \ldots, (x_N, y_N)\}$ with $N$ samples, $r$ numerical features and $c$ categorical features, and $L$ labels, where each $x_i \in \mathbb{R}^r \times \mathbb{C}^c$ and $y_i \in Y = \{1, \ldots, L\}$. Following Cui et al. (2022), we only consider classification tasks in this work, but it should be noted that regression can be easily added into our framework. Note that features may contain missing values. After appropriate preprocessing steps to convert the categorical variables to numerical ones and imputing the missing values, [1] we can directly apply some existing distillation schemes such as KIP (Nguyen et al., 2021) or GM (Zhao et al., 2021). This procedure is sketched in Algorithm 1.

## 2.1 Representation Learning via Column Embedding

A key ingredient in the development of neural networks for tabular data is the use of column embeddings. First developed for categorical features, the idea is to learn an embedding for each of the categories in a categorical feature (Guo & Berkhahn, 2016). This embedding would replace the one-hot encoded numerical representation of the categories and be used in conjunction with the (appropriately scaled and imputed) numerical features in standard and custom feed-forward networks (FFNs) (Borisov et al., 2024). Column embeddings for numerical data were developed to use more standard modern architectures such as graph neural networks (GNNs) and Transformers. As with categorical data, each numerical value in a numerical feature of the table would be converted into a learnable embedding. Thus, more precisely, a sample (row) in a table with $r$ numerical features and $c$ categorical features is now represented as a set of $(r + c)$ embeddings in $\mathbb{R}^m$ each of size $m$ (where $m$ is a user-specified hyperparameter), thus effectively as the $(m \times (r + c))$ matrix. [2]

**Encoder Architectures.** Given the $\mathbb{R}^{m \times (r+c)}$ representation of a row (sample) using column embeddings, our goal is to learn a more compact yet faithful representation of a row. One simple strategy is to concatenate all the $(r + c)$ column embeddings into a single vector in $\mathbb{R}^{m(r+c)}$ of size $m(r + c)$ and input it into an FFN which projects it down to a lower dimensionality (fig. 8). However, one of our main motivations for using column embeddings is to leverage the capabilities of more modern architectures. For a given row, the $(r + c)$ column embeddings can be treated as initial token embeddings that are progressively updated through multiple Transformer blocks as described by Gorishniy et al. (2021). Using a dummy `[CLS]` token, the above process can create a $m$-dimensional representation of the row (fig. 11). An alternate procedure is to represent a table as a bipartite graph between columns and rows (with column values and rows as vertices) and utilize the column embeddings as representations for the column vertices (Wu et al., 2021). Then, the row embeddings are obtained by filling in representations for the row vertices via multiple rounds of message passing in a multi-layered GNN (fig. 9). For our purposes, we consider all three architectures – FFN, Transformer and GNN – as encoders that project the $\mathbb{R}^{m \times (r+c)}$ representation of row into an embedding in $\mathbb{R}^m$. While categorical column embeddings are standard, there are multiple techniques for numerical column embeddings (Gorishniy et al., 2021; 2022). We discuss and ablate the effect these different schemes have in appendix B.2.

**Learning Objective.** Our goal is to retain as much information regarding the original data in the learned representation as possible. *The need for high-fidelity learned representations is critical because we do not assume anything regarding the downstream model, which will be trained with the distilled data*. Thus, we try to reconstruct the original data from the learned representation as well as possible. Formally, given column embeddings $C : \mathbb{R}^r \times \mathbb{C}^c \to \mathbb{R}^{m \times (r+c)}$, and an encoder $\phi : \mathbb{R}^{m \times (r+c)} \to \mathbb{R}^m$, we utilize a decoder $\psi : \mathbb{R}^m \to \mathbb{R}^r \times \mathbb{C}^c$ to reconstruct the original data, and solve the following optimization problem:

$$\min_{C, \phi, \psi} \sum_{(x,y) \in S} \ell\left(x, \psi(\phi(C(x)))\right), \tag{1}$$

where $\ell(\cdot, \cdot)$ is a reconstruction error (RE). Note that the above representation learning does not use the label information in the data $S$. This representation learning framework allows us to infuse class information in the representations while ensuring no loss of original information. Thus, after

---

[1] For example, using data science tools such as `preprocessing.OneHotEncoder` and `impute.SimpleImputer` from the `scikit-learn` machine learning toolkit.

[2] While each feature can have column embeddings of different sizes, many neural network architectures require the column embedding size to match across all features.

obtaining the column embeddings $C$, encoder $\phi$ and decoder $\psi$ by solving eq. (1), we fine-tune the encoder by learning a classifier $f : \mathbb{R}^m \to Y$ on top of the learned representations while keeping the reconstruction loss low:

$$\min_{C,\phi,\psi,f} \sum_{(x,y)\in S} \ell(x, \psi(\phi(C(x))) + \alpha \mathcal{L}(y, f(\phi(C(x))))), \tag{2}$$

where $\mathcal{L}(\cdot, \cdot)$ is the downstream learning loss function, and $\alpha > 0$ is a hyperparameter balancing the classification and reconstruction quality. Appendix A.4.2 discusses this procedure in more detail.

---

**Algorithm 2: `TDColER`**: Distill dataset $S$ with $N$ samples given *distiller* $F : \mathbb{R}^{N\times m} \times Y^N \to \mathbb{R}^{n\times m} \times Y^n$, and learnable *column embeddings* $C : \mathbb{R}^r \times \mathbb{C}^c \to \mathbb{R}^{m\times(r+c)}$, *encoder* $\phi : \mathbb{R}^{m(r+c)} \to \mathbb{R}^m$, *decoder* $\psi : \mathbb{R}^m \to \mathbb{R}^r \times \mathbb{C}^c$, *classifier* $f : \mathbb{R}^m \to Y$.

---

1  $C, \phi, \psi \leftarrow$ solve eq. (1)       // minimize RE
2  $C, \phi, \psi, f \leftarrow$ solve eq. (2)       // fine-tune
3  $\tilde{S} \leftarrow \{(\phi(C(x)), y), (x,y) \in S\}$       // Encode
4  $\tilde{R} \leftarrow F(\tilde{S})$  // Distill in latent space
5  $R \leftarrow \{(\psi(x), y), (x,y) \in \tilde{R}\}$       // Decode
6  **return** $R, \tilde{R}, C, \phi, \psi$

---

**Complete Distillation Pipeline.** After the column embeddings $C$, encoder $\phi$ and decoder $\psi$ are learned (with eq. (1)) and fine-tuned (with eq. (2)), we convert the input features of the whole original dataset (with $N$ samples) into the learned representations in $\mathbb{R}^m$ using $C$ and $\phi$ and apply the aforementioned distillation schemes to this dataset ($N$ samples in $\mathbb{R}^m$) to get $n$ distilled samples in $\mathbb{R}^m$. At this point, we decode the distilled samples into the original representation using $\psi$. This whole pipeline is summarized in Algorithm 2. Note that the distillation with the learned representation in $\mathbb{R}^m$, and the availability of the decoder $\psi$, allows us to have two versions of the distilled data – one in the learned representation ($\tilde{R}$ in Algorithm 2, Line 4), and one in the original representation ($R$ in Algorithm 2, Line 5). We can choose the appropriate distilled set based on the downstream application: If we require the distilled data to be obfuscated with no explicit correspondence to the original features, we can use $\tilde{R}$. In this setting, we are required to have the column embeddings $C$ and the encoder $\phi$ during inference with the downstream trained model to map the test points into the appropriate representation. If we require the distilled data and the model trained on it to be interpretable in terms of the original features, we should use the distilled set $R$ in the original representation. In this case, we do not need the column embeddings or the encoder during inference.

**Remark 1.** *Our contribution is a novel representation learning and distillation pipeline for model-agnostic tabular data distillation utilizing existing distillation schemes, column embeddings, and network architectures such as transformers and GNNs. In our thorough empirical evaluations, we will demonstrate the distilled data quality boost from this pipeline across multiple datasets and downstream models.*

## 3  EVALUATION BENCHMARK

To thoroughly evaluate the various configurations of the proposed distillation pipeline, we establish a comprehensive benchmark suite with a varied set of datasets and downstream models, evaluating the pipeline at various levels of distillation sizes. With 3 encoder architectures, 6 distillation schemes (including variants), 20+ datasets, 7 downstream models, 10 distillation sizes, 5 repetitions per distillation pipeline, and model training, we have generated over 226,200 distilled datasets and trained over 541,980 individual downstream models [3]

**Datasets.** We consider 23 datasets from OpenML (Vanschoren et al., 2013) with the number of samples varying from 10,000 to over 110,000, and number of features varying from 7 to 54. Instead of investigating a few *large* datasets, we choose to incorporate more datasets to generalize the findings across a wider range of datasets. The datasets are chosen to be diverse in terms of the number of samples, features, and the type of features (numerical, categorical, or mixed). There are 14/23 datasets

---

[3]The `TDBench` benchmarking suite (code provided in the supplement) can be extended to evaluate any new distillation method, tabular representation, and downstream model and compared against our current database of results (also provided in the supplement). The API requirements for each of these components in the distillation pipeline are described in appendix C, and the procedure to execute the benchmark suite can be found in appendix C.2, and the comparison using the current database of results can be found in appendix C.1.

with only numerical features, 2/23 with only categorical features, and 7/23 with both numerical and categorical features. All these datasets correspond to binary classification problems. Class imbalance is a common feature of tabular datasets (Johnson & Khoshgoftaar, 2019; Thabtah et al., 2020), and we focus on binary classification to carefully study the effect of class imbalance on the distilled data quality. There are 9/23 almost perfectly balanced datasets and 10/23 datasets with a ratio of close to 1:2 between the smaller and larger classes, with the worst imbalance ratio smaller than 1:15. Note that while we only consider binary classification datasets, the distillation pipelines are natively applicable to multi-class classification problems.

**Distillation Methods.** Given our aforementioned desiderata for model-agnosticity, we have the following existing distillation schemes available, which take as input the set $S$ of $N$ samples and output a set $R$ of $n \ll N$ distilled samples (further details regarding implementation of each distillation method is provided in appendix A.5.2):

- $k$**-means Clustering (KM)** finds $n/L$ clusters for each of the $L$ classes to produce a total of $n$ distilled samples using Lloyd's $k$-means algorithm (Lloyd, 1982). We consider two variations here by (i) using the Euclidean center of each cluster to generate a synthetic sample or (ii) choosing the closest *real* point to the Euclidean center of each cluster. That is, $R$ comprises $n/L$ cluster centers (or closest real points) for each of the $L$ classes.
- **Agglomerative Clustering (AG)** (Müllner, 2011) again generates $n/L$ clusters for each of the $L$ classes is similar to $k$-means. We use the Ward linkage scheme with the Euclidean distance metric. Similar to $k$-means, we generate (i) synthetic samples by using the Euclidean center of a cluster or (ii) real samples that are closest to the cluster centers.
- **Kernel Induced Points (KIP)** (Nguyen et al., 2021) uses the neural tangent kernel (NTK) (Jacot et al., 2018) of a wide neural network and kernel ridge regression to produce a distilled set of samples. Given the feature matrix $X \in \mathbb{R}^{N \times D}$ and the label vector $\mathbf{y} \in Y^N$, KIP learns the distilled feature matrix $\bar{X} \in \mathbb{R}^{n \times D}$ and label vector $\bar{\mathbf{y}} \in Y^n$ by solving the following problem:

$$\min_{\bar{X}, \bar{\mathbf{y}}} \mathcal{L}\left(\mathbf{y}, K_{X\bar{X}}(K_{\bar{X}\bar{X}} + \lambda I)^{-1}\bar{\mathbf{y}}\right), \tag{3}$$

  where $\mathcal{L}$ is the downstream learning loss function, $K_{X\bar{X}} \in \mathbb{R}^{N \times n}$ is the NTK matrix between $X$ and $\bar{X}$, $K_{\bar{X}\bar{X}} \in \mathbb{R}^{n \times n}$ is the NTK matrix of $\bar{X}$ with itself, and $\lambda > 0$ is a regularization hyperparameter for the kernel ridge regression. Essentially, we are learning a set of synthetic samples such that the predictions made on the original dataset features using the distilled dataset via kernel ridge regression match the original labels.
- **Gradient Matching (GM)** (Zhao et al., 2021) produces the distilled set $R$ for a given "backbone model" $M_\theta$ (parameterized by $\theta$) by directly optimizing for $R$ to induce model parameter gradients that are similar to the gradients obtained while training $M_\theta$ on the full dataset $S$. Given a distance metric $D(\cdot, \cdot)$, and a distribution $P_{\theta_0}$ over the random model parameter initializations $\theta_0$, the distillation problem tries to minimize the distance between the model gradients computed on the full and distilled datasets over the $T$ steps of model learning as follows:

$$\min_R \mathbb{E}_{\theta_0 \sim P_{\theta_0}} \left[ \sum_{t=0}^{T-1} D\left(\nabla_\theta \mathcal{L}(\theta_t; S), \nabla_\theta \mathcal{L}(\theta_t; R)\right) \right], \tag{4}$$

  where $\mathcal{L}(\theta; S)$ is the loss of the model $M_\theta$ on the original full dataset $S$, $\mathcal{L}(\theta; R)$ is the loss of $M_\theta$ evaluated on $R$, and the model parameters $\theta_t$ are updated at $\theta_{t+1} \leftarrow \theta_t - \eta_\theta \nabla_\theta \mathcal{L}(\theta_t; S)$ via gradient descent with a learning rate $\eta_\theta$ using the full original dataset.

We consider KIP and GM as representatives from previous data distillation literature that are *model-agnostic* and *model-centric*, respectively. Appendix A.5.1 further discusses our choice of distillation methods considered in this work. All the above distillation schemes require the data to be preprocessed into a numerical form, and can be used in Algorithm 1 to distill tables. But, as we will see, this is not a very useful scheme. Our evaluation of **TDColER** on TDBench will demonstrate how the performance of these distillation schemes are boosted via representation learning.

To study the ability of the distillation pipeline to generate really small but useful distilled datasets, we consider extremely small distilled datasets with 10-100 instances per class (IPC), corresponding to a distillation fraction of the order of 0.1-1.0% on the smallest datasets, and 0.01-0.1% for the largest datasets. This is comparable to the compression ratio of 0.02-1% used in Cui et al. (2022) and Cazenavette et al. (2023).

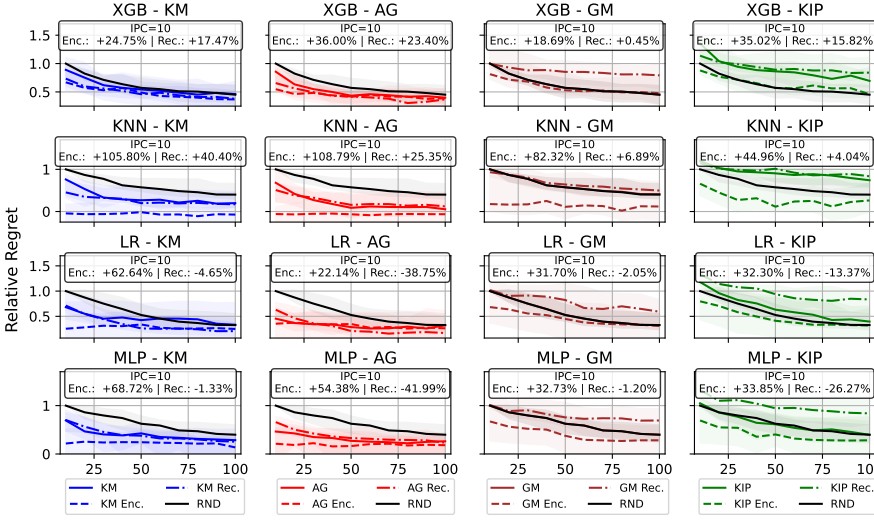

Figure 3: Change in relative regret of downstream classifiers when trained on distilled data over IPC $\in [10, 100]$, aggregated over datasets and encoder architectures. Lower is better. Each column corresponds to a downstream classifier, and each row represents a representation scheme – original, encoded (Enc.), and reconstructed (Rec.). Data distilled by clustering methods (AG, KM) in the encoded space show the best performance for all classifiers. In many cases, using the encoded representation as the final output yields a performance comparable to using the original representation. Figure 12 shows a more detailed version of this plot that includes FTTransformer and ResNet.

**Downstream models.** We consider 7 downstream models to evaluate the distilled data quality. We consider the Nearest-Neighbor Classifier (KNeighbors), Logistic Regression (LR), Gaussian Naive Bayes (GNB), and the Multi-Layered Perceptron (MLP) from the `scikit-learn` library (Pedregosa et al., 2011). We also consider the popular XGBoost ensemble of gradient-boosted decision trees (XGB) (Chen & Guestrin, 2016). We include two recent neural network models for tabular data, the ResNet and the FTTransformer models (Gorishniy et al., 2021). Since our distillation pipeline is deliberately model-agnostic, we train these models on the distilled data using the default hyperparameters of the corresponding libraries. We also consider a hyperparameter optimization (HPO) use case using the distilled datasets in our evaluations, which can be found in section 4.

**Evaluation metric.** To have a standardized way to quantify the quality of the distilled data across different models and datasets, we use the notion of *relative regret* which compares the model's balanced accuracy score when trained on the full, distilled and randomly sampled data points. Precisely, the *relative regret* is defined as $(A_F - A)/(A_F - A_{R_{10}})$, where $A_F$ is the balanced accuracy of the model trained on the full training set, $A_{R_{10}}$ is the balanced accuracy on the same test set when trained on 10 random samples per class averaged over 5 random repetitions, and $A$ is the balanced accuracy of the model when trained on the distilled dataset over random 5 repetitions. A relative regret of 1 matches the performance of random sampling at IPC=10, and a relative regret of 0 matches the performance of the model trained on the full dataset (which is usually the gold standard) – lower relative regret implies higher distilled data quality [4].

## 4 RESULTS ANALYSIS

In this section, we present the analysis of the results obtained from our benchmarking experiments. For the sake of brevity, we will use the following acronyms – Instances Per Class: IPC, $k$-means: KM, agglomerative: AG, gradient matching: GM, kernel inducing points: KIP, feed-forward neural network: FFN, graph neural network: GNN, transformer: TF. Additionally, the supervised-fine-tuned variant of the autoencoder will be marked with a *. For example, the results of Algorithm 2 with a

---

[4]For all the downstream models, the aggregate (median across all datasets) relative regret of random samples at IPC=10 (smallest distillation size) is 1.0 by definition, while the aggregate relative regret of random samples at IPC=100 (largest distillation size) is around 0.5, indicating that the benchmark is challenging enough with significant room for improvement.

transformer architecture for $\phi$ as *TF\**, whereas *TF* denotes the version that skips line 2 of Algorithm 2 to highlight the importance of the supervised fine-tuning.

**How beneficial are the learned representations for distillation?** As the first step of our analysis, we examine the performance difference between pipelines that use encoder's latent space and those that do not. To fully understand the effect of our latent space projection step, we analyze our results from two angles: 1) Is it better to distill in the latent or original space? 2) If latent space is better, is it better to decode the data back to the original space or stay in the latent space?

Table 1: Average rank and median relative regret of distillation pipelines that use the latent space of different encoder architectures evaluated at IPC=10, grouped over all datasets and classifiers.

| Encoder | Mean Rank | Median R.R. |
|---------|-----------|-------------|
| TF      | 4.1176    | 0.9439      |
| FFN     | 4.3407    | 0.9746      |
| GNN     | 4.2243    | 0.9695      |
| TF*     | **2.3591**| **0.6149**  |
| FFN*    | 3.3652    | 0.8082      |
| GNN*    | 2.5931 | 0.7135 |

Figure 3 shows the relative regret score of distillation methods under different data representation schemes. We start by examining the downstream performance difference between pipelines that use the latent space to distill in vs. ones that do not (Algorithm 2 vs. Algorithm 1). The results show that using the latent space is highly beneficial in most cases with lower IPC values. This trend is most apparent in classifiers such as KNN (44.96-108.79% improvement at IPC=10), Logistic Regression (22.14-62.64% improvement) or MLP (32.73-68.72% improvement), while XGBoost shows the least improvement from any of the distillation methods (15.82-36.00% improvement). $k$-means and agglomerative clustering also show a more apparent decrease in regret, while KIP and GM show noticeable improvements only when both the distillation and the final dataset are in the latent space. With this in mind, we examine the performance difference when training on the distilled data in the latent space or decoding to the original space before training the downstream classifier (using $\tilde{R}$ or $R$ from Algorithm 2). Figure 3 shows that training on the dataset in the latent space improves the downstream performance for all distillation pipelines – in fact, it is the best performer for almost every instance over classifiers and distillation methods. The change in performance is more apparent in KNN (40.92-65.40%), Logistic Regression (33.75-67.29%) and MLP (33.93-96.38%), while XGBoost shows a more subtle change (7.28-19.20%). This leads us to conclude that **distillation methods benefit the most when both distilling *and* downstream training in on the latent representations**. It is also worth noting that decoding the distilled data from the latent space (Rec.) is also beneficial compared to random sampling in many cases.

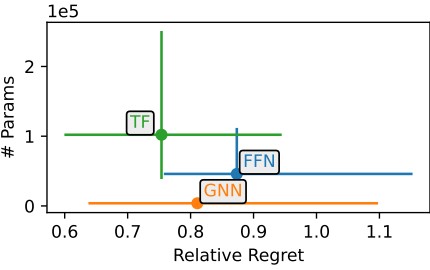

Figure 4: Scatterplot of encoder parameter size and downstream classifier regret at IPC=10 aggregated over datasets and classifiers. The dots represent the median values, and the error bars span the 25% and 75% percentile, respectively. Note that the encoder sizes for both SFT and base versions are the same for each dataset.

**How do different encoders compare?** Having observed that using the latent space is beneficial, we now seek to identify which encoder architecture leads to the best performance. Table 1 shows the average rank of distillation pipelines that use the latent space of different encoder architectures. Among the tested architectures and training objectives, the **transformer architecture with supervised fine-tuning** leads to the best downstream performance. We find that **adding supervised fine-tuning improves the downstream performance of all encoders** in general.

Another important aspect of data distillation is to improve downstream classifier efficiency providing a lightweight proxy. Thus, it is important to examine the resources required in the distillation pipeline. Specifically, one aspect of our distillation pipelines that can add an additional cost is the encoder. In settings that require the data to be projected into latent space at inference time, the encoder can be considered part of the distilled data. Figure 4 shows the parameter size of the different encoder architectures vs. the downstream classifier regret scores. As noted before, the transformer architecture leads to the best downstream performance. However, it is worth noting that *GNN architecture has the*

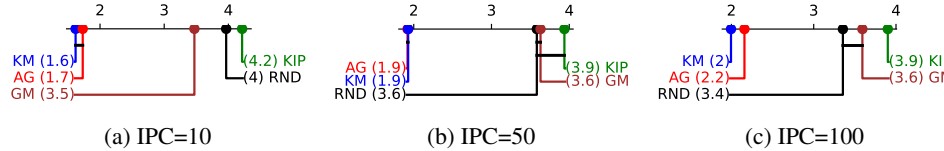

(a) IPC=10          (b) IPC=50          (c) IPC=100

Figure 5: Critical difference plot comparing ranks of distillation methods across datasets, encoders, and classifiers per IPC value. The x-axis denotes the average rank, and a black horizontal line connects groups of methods that are *not significantly different* in the rank distribution. $k$-means and agglomerative are indistinguishable from each other in IPC $\in \{10, 50\}$, but $k$-means gains an edge in IPC=100.

Table 3: Relative regret of pipelines that use different combinations of distill methods and encoders at IPC=10, aggregated over classifiers. The best value for each column is marked with **bold**, and the second best is marked with underline. The best in each distillation method group is marked with *italics*. On average, $k$-means with SFT transformer shows the best performance, but agglomerative clustering also shows comparable performance.

| Distill Method | Encoder | Regret | | | | | |
| --- | --- | --- | --- | --- | --- | --- | --- |
| | | Min | Q1 | Mean | Median | Q3 | Max |
| KM | TF* | *-14.4491* | ***0.0733*** | ***-0.0464*** | ***0.4056*** | 0.7379 | 1.1773 |
| | FFN* | -11.9912 | 0.2039 | 0.1382 | 0.6035 | 0.8389 | 1.5368 |
| | GNN* | -12.1045 | 0.0973 | 0.1054 | 0.5047 | 0.7887 | ***1.0494*** |
| AG | TF* | ***-15.3965*** | *0.0810* | *0.0187* | *0.4135* | ***0.6507*** | *1.4982* |
| | FFN* | -10.1288 | 0.2483 | 0.3695 | 0.6230 | 0.8823 | 4.1191 |
| | GNN* | -13.1881 | 0.1397 | 0.2245 | 0.4793 | 0.7595 | 4.4801 |
| KIP | TF* | -4.1619 | *0.5226* | *1.1124* | *0.9415* | *1.2966* | 11.1034 |
| | FFN* | *-5.3973* | 0.8053 | 1.6363 | 1.2502 | 1.6434 | 16.4137 |
| | GNN* | -1.4649 | 0.7403 | 1.1957 | 1.0136 | 1.3329 | *10.5175* |
| GM | TF* | -3.8002 | *0.4105* | *0.7273* | *0.7952* | 1.0564 | *4.9450* |
| | FFN* | *-4.3269* | 0.5975 | 1.2660 | 0.9938 | 1.3827 | 16.5044 |
| | GNN* | -1.4776 | 0.4626 | 0.8073 | 0.8457 | *0.9779* | 8.4566 |

*smallest overall parameter size while providing the second-best performance.* Further discussion on the parameter size analysis of each encoder architecture can be found in appendix A.3.

**Which distillation method leads to the best downstream performance?** We now compare the most critical piece of the distillation pipeline – the distillation method. We wish to understand which method leads to the best downstream performance across datasets, encoders, and classifier configurations. To evaluate, we perform a Wilcoxon signed-rank test to identify groups that stand out from the rest, as shown in fig. 5. The results show that **clustering-based methods ($k$-means, agglomerative) show the strongest performance across datasets and encoder configurations**, consistently placing in the top two ranks. While both methods show similar performance, we find that $k$-means starts to outperform agglomerative as the IPC increases.

Table 2: The best performers of each dataset are classifiers ranked by their appearance count at the top 3 of each comparison at IPC=10. $k$-means stands out as the strongest performer in combination with a supervised-fine-tuned transformer encoder.

| Count | Encoder | D.M. | Output |
| --- | --- | --- | --- |
| 67 | TF* | KM | Enc. |
| 63 | GNN* | KM | Enc. |
| 61 | GNN* | AG | Enc. |
| 61 | TF* | AG | Enc. |
| 42 | FFN* | KM | Enc. |

**Which combination leads to best performance?** Our previous analysis has revealed that transformer encoders with SFT and clustering-based distillation methods perform best in their respective comparisons. Now, we aim to identify which combination of encoder and distillation method leads to the best downstream performance. We approach this question by examining the detailed statistics behind the combinations' performance and the top performers of each dataset, classifier, and $n$ combinations. Table 3 shows detailed statistics about each distill method and encoder combination, while

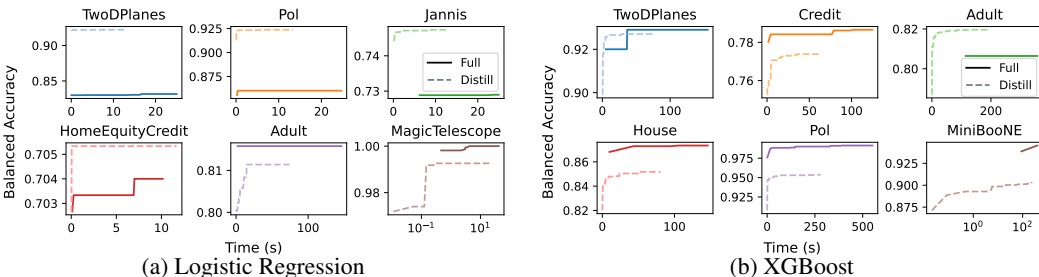

Figure 6: Comparison of HPO validation performance vs. runtime when using full and distilled data. To better visualize the performance difference, we truncate the plot for the full data run at twice the runtime of the entire distilled run.

table 2 shows the count of the top 5 distillation pipelines that placed in the top 10 by performance in each comparison group. In line with our previous findings, the results show that $k$-**means clustering with supervised-fine-tuned transformer encoder leads to the best overall performance**. All of the top performers are clustering-based methods, and all of them use the latent space, again confirming that **using the latent representation from the encoder greatly benefits distillation methods**. In addition, the GNN encoder shows a comparative performance to that of the transformer encoder. This is especially noteworthy, considering that GNN has the smallest parameter size among the encoder architectures.

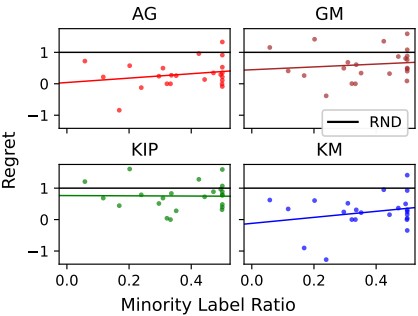

Figure 7: Average median relative regret of distillation methods aggregated over downstream classifiers and encoders at IPC=10 with a least-squares linear regression. Compared to KIP and GM, $k$-means and agglomerative show much stronger performance in imbalanced data.

As mentioned in section 3, we additionally run a smaller-scale HPO experiment to consider a use case for distilled data, as seen in fig. 6. Specifically, we consider a case where the validation and testing data is sampled from the original data, and the classifier is trained on the full or distilled data. In general, we note that training on the distilled data gives comparable performance to training on the full data in a fraction of the time, consuming on average 21.84% of the runtime and reaching 98.37% of the performance.

**How does class imbalance affect performance?** Finally, we examine the downstream performance of classifiers with respect to the label balance, or the imbalance, of the original dataset, shown fig. 7. Compared to other methods, including random sampling, clustering-based methods show impressive strength when distilling datasets with high label imbalance, highlighting their robustness under challenging data distributions. One possible explanation behind this phenomenon is that while NN-based distillation methods may prioritize the majority class due to the imbalance, the clustering methods are forced to place equal emphasis on all classes, preventing an overfitting on the majority class.

## 5 DISCUSSION

This work introduced a tabular data distillation pipeline and evaluated it extensively leveraging various distillation methods, with a focus on supporting both non-NN and NN ML classifiers. We introduce a novel framework, `TDColER`, that leverages latent representation of tabular data in distillation, and evaluate it thoroughly in our benchmark, `TDBench`, which included 23 datasets, 4 distillation algorithms, 3 autoencoder architectures, and 7 downstream classifiers, resulting in over 226,200 distilled datasets and 541,980 downstream classifier instances. Our results show that `TDColER` can induce superior performance in distillation methods on tabular data, improving the quality by 0.5-143%. We also show that $k$-means clustering and transformer autoencoder are a particularly strong combination for tabular data distillation. We hope that this work will serve as a starting point for future research in tabular data distillation and plan to extend this benchmark further to incorporate new distillation pipelines.

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

Table 4: Dataset name and OpenML Vanschoren et al. (2013) url

| Dataset Name | Dataset URL |
|---|---|
| adult | https://api.openml.org/d/1590 |
| Amazon_employee_access | https://api.openml.org/d/4135 |
| Bank_marketing_data_set_UCI | https://api.openml.org/d/44234 |
| credit | https://api.openml.org/d/45027 |
| default-of-credit-card-clients | https://api.openml.org/d/45020 |
| Diabetes130US | https://api.openml.org/d/45022 |
| electrcity | https://api.openml.org/d/151 |
| elevators | https://api.openml.org/d/846 |
| higgs | https://api.openml.org/d/23512 |
| hcdr | https://api.openml.org/d/45071 |
| house_16H | https://api.openml.org/d/821 |
| jannis | https://api.openml.org/d/45021 |
| law-school-admission-bianry | https://api.openml.org/d/43890 |
| MagicTelescope | https://api.openml.org/d/1120 |
| Medical-Appointment-No-Shows | https://api.openml.org/d/43439 |
| MiniBooNE | https://api.openml.org/d/44088 |
| numerai28.6 | https://api.openml.org/d/23517 |
| nursery | https://api.openml.org/d/959 |
| PhishingWebsites | https://api.openml.org/d/4534 |
| pol | https://api.openml.org/d/722 |
| road-safety | https://api.openml.org/d/44161 |
| Click_prediction_small | https://api.openml.org/d/1220 |
| 2dplanes | https://api.openml.org/d/727 |

Table 5: Metadata of each dataset seen in table 4

| Dataset | # Instances | # Features | # Cont. | # Cat. | # Class 0 | # Class 1 |
|---|---|---|---|---|---|---|
| 2dplanes | 40,768 | 10 | 10 | 0 | 20,420 | 20,348 |
| Amazon_employee_access | 32,769 | 9 | 8 | 1 | 1,897 | 30,872 |
| Bank_marketing_data_set_UCI | 45,211 | 16 | 7 | 9 | 39,922 | 5,289 |
| Click_prediction_small | 39,948 | 11 | 11 | 0 | 33,220 | 6,728 |
| Diabetes130US | 71,090 | 7 | 7 | 0 | 35,545 | 35,545 |
| MagicTelescope | 19,020 | 11 | 11 | 0 | 12,332 | 6,688 |
| Medical-Appointment-No-Shows | 110,527 | 13 | 10 | 3 | 88,208 | 22,319 |
| MiniBooNE | 72,998 | 50 | 50 | 0 | 36,499 | 36,499 |
| PhishingWebsites | 11,055 | 30 | 0 | 30 | 4,898 | 6,157 |
| adult | 48,842 | 14 | 6 | 8 | 37,155 | 11,687 |
| credit | 16,714 | 10 | 10 | 0 | 8,357 | 8,357 |
| default-of-credit-card-clients | 13,272 | 20 | 20 | 0 | 6,636 | 6,636 |
| electrcity | 45,312 | 8 | 7 | 1 | 26,075 | 19,237 |
| elevators | 16,599 | 18 | 18 | 0 | 5,130 | 11,469 |
| hcdr | 10,000 | 22 | 22 | 0 | 5,000 | 5,000 |
| higgs | 98,050 | 28 | 28 | 0 | 46,223 | 51,827 |
| house_16H | 22,784 | 16 | 16 | 0 | 6,744 | 16,040 |
| jannis | 57,580 | 54 | 54 | 0 | 28,790 | 28,790 |
| law-school-admission-bianry | 20,800 | 11 | 6 | 5 | 6,694 | 14,106 |
| numerai28.6 | 96,320 | 21 | 21 | 0 | 47,662 | 48,658 |
| nursery | 12,960 | 8 | 0 | 8 | 8,640 | 4,320 |
| pol | 15,000 | 48 | 48 | 0 | 5,041 | 9,959 |
| road-safety | 111,762 | 32 | 29 | 3 | 55,881 | 55,881 |

# A    APPENDIX

## A.1    DATASETS

Tables 4 and 5 show the information about datasets used in our experiments along with their OpenML Vanschoren et al. (2013) URLs.

Table 6: Hyperparameters tested for FFN encoder.

| Hyperparameter | Values |
|---|---|
| d_hidden | $(100, 200)$ |
| n_hidden | $[1, 4]$ |
| dropout | $(0, 0.2, 0.4)$ |
| d_embedding | $(10, 20, 50, 100, 200)$ |
| use_embedding | (True,False) |
| learning_rate | $10^{[-3,-1]}$ |
| weight_decay | $10^{[-4,-1]}$ |
| lr_scheduler | (None, Plateau, Cosine) |

Table 7: Hyperparameters tested for GNN encoder.

| Hyperparameter | Values |
|---|---|
| graph_layer | (graphsage, gcn, gat) |
| graph_aggr | (mean, softmax) |
| n_graph | $[1, 15]$ |
| edge_direction | (bidirectional, multipass) |
| edge_dropout | $(0, 0.2, 0.4)$ |
| learning_rate | $10^{[-3,-1]}$ |
| weight_decay | $10^{[-4,-1]}$ |
| lr_scheduler | (None, Plateau, Cosine) |

## A.2 HYPERPARAMETER OPTIMIZATION FOR ENCODERS

Tables 6 to 10 show the hyperparameters considered for different modules of the autoencoders. We use $\{x, y, z\}$ to denote a set of variables and $[a, b]$ to denote an inclusive range of values. We conduct HPO for each autoencoder + dataset pair using an implementation of hyperopt Bergstra et al. (2015) from Ray Tune Liaw et al. (2018) with a maximum of 500 samples for each HPO run. As noted in section 2.1, we first train the vanilla autoencoders for each dataset using the encoder hyperparameters seen in tables 6 to 8 and decoder parameters seen in table 9. Once the vanilla autoencoders are trained, we then conduct an additional fine-tuning with a classifier head with hyperperameters seen in table 10 where $\alpha$ is used to balance the objective functions of the decoder and classifier heads.

Table 8: Hyperparameters tested for TF autoencoder.

| Hyperparameter | Values |
|---|---|
| n_blocks | $[1, 10]$ |
| n_attention_heads | $2^{[1,4]}$ |
| d_qkv | $2^{[0,7]}$ |
| layer_norm_eps | $10^{[-5,-1]}$ |
| d_mlp | $2^{[7,11]}$ |
| d_mlp_hidden | $(100, 200)$ |
| n_mlp_hidden | $[1, 4]$ |
| dropout | $[0, 0.4]$ |
| learning_rate | $10^{[-3,-1]}$ |
| weight_decay | $10^{[-4,-1]}$ |
| lr_scheduler | (None, Plateau, Cosine) |

Table 9: Hyperparameters tested for decoders. The decoder architecture is kept the same for all encoders and optimized individually.

| Hyperparameter | Values |
| --- | --- |
| d_hidden | $(100, 200)$ |
| n_hidden | $[1, 4]$ |
| learning_rate | $10^{[-3,-1]}$ |
| weight_decay | $10^{[-4,-1]}$ |
| lr_scheduler | $(\text{None}, \text{Plateau}, \text{Cosine})$ |

### A.3   DISCUSSION ON PARAMETER SIZE OF AUTOENCODERS

Here we expand on our parameter size of the encoder architectures of the autoencoders. This is worth noting because if the distilled data is in the latent space, the encoder module is required to project any new data to the same space. Thus, the encoder is considered to be a part of the distilled output.

We can characterize the parameter size of each encoder architecture given a $D$-dimensional binarized dataset with $c$ categorical features and $r$ continuous features that is projected to a $d$-dimensional latent space.

**FFN.** We used an FFN architecture with an $M$-dimensional embedding layer followed by $H$ hidden layers that receive and output $W$-dimensional vectors. The parameter size of such an FFN is as follows:

$$O(DM + (c + r)MW + HW^2 + Wd) \tag{5}$$

The column embeddings are of size $O(DM)$, the input layer maps the concatenated $(c + r)M$-dimensional vector to hidden layer dimension $W$ with $(c + r)MW$ size. The hidden layers are of sizes $O(W^2)$ each for $H$ hidden layers. The output layer maps the $W$-dimensional hidden layer output to the desired $d$-dimensions.

**GNN.** We use a GNN encoder with $H$ consecutive layers. The dimension of the vectors passed between the graph layers are fixed to $d$, meaning that $M = d$. Thus, each graph layer maintains a $d$ by $d$ matrix to handle a $d$-dimensional input vector and output a $d$-dimensional vector.

$$O(Dd + Hd^2) \tag{6}$$

The column embeddings are of size $O(Dd)$ since $M = d$. Each of the $H$ GNN layers is of size $O(d^2)$.

**Transformer.** We consider an implementation of a transformer autoencoder inspired by the architecture of FT-Transformer Gorishniy et al. (2021). The encoder has an $M$-dimensional embedding layer followed $H$ transformer blocks. Each transformer block takes in a sequence of $M$-dimensional embeddings and oututs a single $d$-dimensional vector. The block is composed of a multihead-attention module with $m$ heads and a FFN module to project the attention scores back to the input space. We modify the architecture seen in (Gorishniy et al., 2021) by allowing the dimension of the attention head to be configurable – i.e. instead of using $M/m$ as the dimension of a single attention head, we allow the module to compute the attention in $d_{qkv}$. This choice is motivated by the fact that our encoders were trained with a latent size of 16, which may not be wide enough for the TF encoder. We then project the resulting embedding in $d_{qkv}m$-dimension back to $M$-dimensionals with $W_o$.

Table 10: Hyperparameters tested for classifier head in SFT.

| Hyperparameter | Values |
| --- | --- |
| d_hidden | $\{100, 200\}$ |
| n_hidden | $[1, 3]$ |
| dropout | $\{0, 0.2, 0.4\}$ |
| alpha | $\{0.3, 0.5, 0.7\}$ |
| learning_rate | $10^{[-3,-1]}$ |
| weight_decay | $10^{[-4,-1]}$ |
| lr_scheduler | $(\text{None}, \text{Plateau}, \text{Cosine})$ |

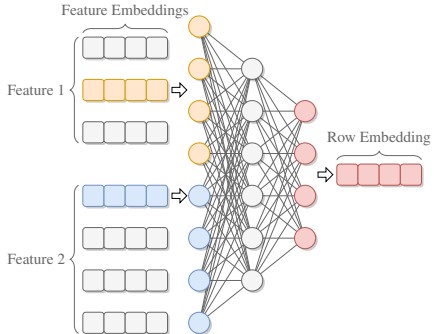

Figure 8: FFN encoder.

Figure 9: GNN encoder.

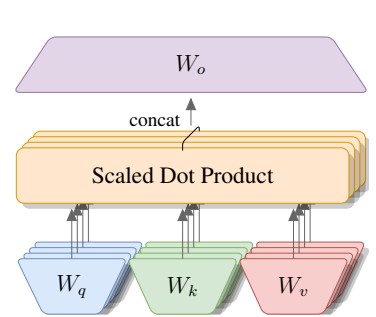

Figure 10: Modified MHA component.

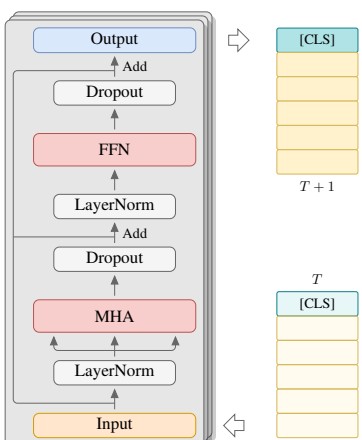

Figure 11: TF encoder block.

Thus, each of $W_q$, $W_k$, $W_v$ and $W_o$ has $d_{qkv}mM$ parameters. The MHA module is then followed by an FFN module which takes a $M$-dimensional vector and projects it back to $M$-dimensions with a $W$-dimensional hidden layer.

$$O(H(4d_{qkv}Mm + 2MW)) \tag{7}$$

### A.4 AUTOENCODER IMPLEMENTATION DETAILS

#### A.4.1 OPTIMIZATION FUNCTION

For the decoder $\psi : \mathbb{R}^d \to \mathbb{R}^D$, we consider a multi-layered fully-connected feed-forward network. Given the encoder $\phi$ and the decoder $\psi$, we use a group-wise softmax operator $\sigma$ to map the output of the decoder to a per-input-feature probability simplex: given an initial binary vector $b \in \{0,1\}^D$ constituting per-input-feature one-hot encodings $b^i$ (that is $b = [b^1 \oplus \ldots \oplus b^{c+r}]$), and a decoder output $B \in \mathbb{R}^D$ with per-input-feature constituents $B^i$ (that is $B = [B^1 \oplus \ldots \oplus B^{c+r}]$, we apply the softmax operation to each per-input-feature constituent to get $\hat{b} = [\hat{b}^1 \oplus \ldots \oplus \hat{b}^{c+r}] \in [0,1]^D$, where $\hat{b}^i = \mathsf{softmax}(B^i)$. We utilize the following per-sample reconstruction loss:

$$\ell(b, \hat{b}) = \frac{1}{c+r} \sum_{i=1}^{c+r} \frac{1}{\log_2 |b^i|} \mathsf{CE}(b^i, \hat{b}^i), \tag{8}$$

where $\mathsf{CE}$ is the standard cross-entropy loss between a one-hot vector and a softmax output, and $|b^i|$ is the length of the $i$-th constituent one-hot encoding in $b$, corresponding to the number of categories (or bins) in the $i$-th categorical (or numerical) feature. This loss is a weighted average of the per-input-feature cross-entropy loss, with weights $(1/\log_2 |b^i|)$ to normalize the loss across all features with varying number of categories or bins.

The encoder and decoder are then learned by optimizing the following unsupervised loss:

$$\mathcal{L}_R(\phi, \psi) = \frac{1}{N} \sum_{(x,y) \in S} \ell\left(P(x), \sigma(\psi(\phi(P(x))))\right), \tag{9}$$

where $P$ is the data homogenizer, and $\sigma$ is the aforementioned group-wise softmax operator. Learning the latent representation in such an unsupervised manner makes this distillation pipeline agnostic to the choice of downstream model. Another advantage of this choice is that the decoder allows us to map the distilled artificial samples in the latent space to the original features, which might be necessary in some applications (for interpretability reasons).

### A.4.2 SUPERVISED LATENT SPACE FINE-TUNING

Given the already learned encoder and decoder, we consider a supervised fine-tuning (FT) step where we utilize a classifier $f : \mathbb{R}^d \to Y$ that utilizes the latent representation. The classifier is learned, and the encoder and decoder are fine-tuned by minimizing the following loss to ensure that the latent space is quite predictive while the reconstruction loss stays low:

$$\mathcal{L}_R(\phi, \psi) + \frac{\alpha}{N} \sum_{(x,y) \in S} \mathsf{CE}(y, f(\phi(P(x)))), \tag{10}$$

where $\alpha > 0$ is penalty parameter to balance the two losses, and $\mathsf{CE}$ is the cross-entropy loss. We consider multi-layer FFN architecture as the classifier $f$.

### A.4.3 ENCODER ARCHITECTURES

**Fully-connected feed-forward network (FFN).** This encoder first selects the column embeddings corresponding to nonzero entries in the binary representation $b$, concatenates them to get a $(c + r)M$-dimensional dense vectors (recall that $b$ will only have $c + r$ nonzeros out of the $D$ dimensions), and inputs them to a fully-connected feed-forward network $\mu : \mathbb{R}^{(c+r)M} \to \mathbb{R}^d$. The encoder $\phi : \{0, 1\}^D \to \mathbb{R}^d$ can be written as:

$$z = \phi(b) = \mu(\oplus([w_i, i \in \{1, \ldots, D\} : b[i] = 1])), \tag{11}$$

where $b[i]$ is the $i$-th entry of the $D$-dimensional vector, and $\oplus$ is the concatenation operator. The FFN $\mu$ and the column embeddings $\{w_i, i \in \{1, \ldots, D\}\}$ constitute the parameters of the encoder $\phi$. For a FFN with $H$ hidden layers, each of width $W$, the total number of parameters in this encoder is $O(DM + (c+r)MW + HW^2 + Wd)$. Figure 8 shows a simplified architecture of the FFN encoder.

**Graph neural network (GNN) encoder.** We also consider a more recent encoder for tabular data proposed in Wu et al. (2021). A bipartite graph is constructed between the column embeddings $\{w_i, i \in \{1, \ldots, D\}\}$ and the (zero-initialized) row (sample) embeddings $\{z_j \in \mathbb{R}^d, j \in \{1, \ldots, N\}\}$, with a bidirectional edge between $w_i$ and $z_j$ if the $b_j[i] = 1$, where $b_j \in \{0, 1\}^D$ is the binary representation of the $j$-th sample. Given the (learned) column embeddings, the row embeddings are obtained via multiple rounds of message passing through multiple GNN layers. This can be written as:

$$\begin{aligned} z_j^h &= \mu_h(z_j^{h-1}, \mathsf{Agg}(w_i^{h-1}, i \in \mathcal{N}_j)), \\ w_i^h &= \mu_h(w_i^{h-1}, \mathsf{Agg}(z_j^h, j \in \mathcal{N}_i)), \end{aligned} \tag{12}$$

where $\mu_h$ is the $h$-th GNN layer, $\mathsf{Agg}$ is an aggregation, $\mathcal{N}_i$ (or $\mathcal{N}_j$) is the neighbor set of the $i$-th column embedding (or $j$-th row embedding). We set the initial $z_j^0 = 0$ (zero-initialized row embeddings), $w_i^0 = w_i$, and utilize $z_j^H$ as the latent representation for distillation after $H$ GNN layers. While Wu et al. (2021) only considered Graph Convolutional Networks Kipf & Welling (2016) as GNN modules, we extend it to GraphSage Hamilton et al. (2017) and Graph Attention Networks Veličković et al. (2018). An important aspect of the GNN encoder is that the desired row embedding size $d$ must match the column embedding size $M$, thus $d = M$. With $H$ GNN layers, the total number of parameters in this encoder is usually $O(Dd + Hd^2)$, which can be significantly smaller than the FFN encoder with moderately sized FFN (large enough $M, W$). Figure 9 shows the graph formulation (left) and the GNN encoder architeture (right).

**Transformer encoder.** Finally, we consider a transformer-based autoencoder inspired by the architecture of FT-Transformer Gorishniy et al. (2021). This encoder uses the same embedding layer as the FFN encoder, which is then followed by transformer blocks. We learn an additional $cls$ embedding, which is placed before all other tokens in every sequence. Each block takes in a sequence (one row) of $d$ embeddings, and is composed of a multihead-attention (MHA) module and a feed-forward network (FFN) module.

For a MHA module with $m$ attention heads, we modify the architecture seen in (Gorishniy et al., 2021) by allowing the dimension of the attention head to be separately configurable – i.e. instead of

Table 11: Parameters of distillation methods.

| Method | Hyperparameter | Value | Description |
|---|---|---|---|
| Common | `distill space` | - | Whether to use the encoder latent space or the raw binary representation. |
| | `use_closest*` | - | Whether to use *median* points instead of the euclidean center. Only applicable to clustering methods. |
| | `output_space`[†] | - | Whether to keep the encoder latent/ decode or use the raw binary space. The binary space is only applicable to clustering methods when `use_closest` is set to True. |
| | `random_seed`[‡] | - | Random seed for distillation algorithm. Not applicable to agglomerative. |
| KIP | `n_epochs` | 1000 | Number of epochs to train the *distilled data*. |
| | `mlp_dim` | 1024 | Width of the neural network to compute the NTK of. |
| GM | `n_epochs` | 500 | Number of epochs to train the *distilled data*. |
| | `mlp_dim` | 1024 | Size of the hidden layer of the target model. |
| | `n_layers` | 2 | Number of hidden layers in the target model. |
| | `lr_mlp` | 0.01 | Learning rate for the target model. |
| | `lr_data` | 0.1 | Learning rate for the *distilled data*. |
| | `mom_data` | 0.5 | Momentum for *distilled data*. |

using $d/m$ as the dimension of a single attention head, we allow the module to compute the attention in $d_{qkv}$. This choice is motivated by the fact that our encoders were trained with a latent size of 16, which may not be wide enough for the TF encoder. We then project the resulting embedding in $d_{qkv}m$-dimension back to $d$-dimension with $W_o$. For an input $w_i$ at the $i$th transformer block, the computation for the MHA module is as follows:

$$a_i = W_o^i(\mathsf{softmax}(\frac{W_q^i(w_i)W_k^i(w_i)}{\sqrt{d_{qkv}}})W_v^i(w_i))$$ (13)

The resulting attention score $a_i$ is then added with the original embedding and passed through an FFN module. Similarly to Gorishniy et al. (2021), the `[cls]` embedding is used as the final output of the encoder. Figure 10 shows our modified MHA component, and fig. 11 shows the TF encoder block.

## A.5  DISTILL METHODS

### A.5.1  CHOICE OF DISTILL METHODS (KIP, GM)

The clustering-based distillation schemes and KIP are not explicitly tied to a specific model and thus satisfy our desiderata of model-agnosticity. In contrast, the Gradient Matching or GM distillation scheme heavily relies on the choice of the backbone model $M_\theta$ (as well as the learning algorithm parameters such as the learning rate), and there is no guarantee that the distilled samples $R$ would be useful for any other model. Thus, this scheme is not model-agnostic. However, we consider GM to be representative of the model-specific distillation schemes for the sake of completeness of our evaluations. For our table distillation, we choose $M_\theta$ to be a multi-layered perceptron with a single hidden layer. This will pose a mismatch when we evaluate the quality of the distilled data $R$ on standard tabular models such as decision tree ensembles and nearest-neighbor models, highlighting the need for model-agnosticity in tabular data distillation.

### A.5.2  DISTILL METHOD IMPLEMENTAION

$k$-**means**   We use the `sklearn.cluster.KMeans` from Pedregosa et al. (2011) with the `n_init` set to `"auto"`.

**Agglomerative**   We use `sklearn.cluster.AgglomerativeClustering` from Pedregosa et al. (2011) with the `linkage` set to `"ward"`. Because agglomerative clustering

Table 12: Hyperparameters of downstream classifiers.

| Classifier | Hyperparameter | Value |
|---|---|---|
| FT-Transformer | d_token | 128 |
| | n_blocks | 2 |
| | attention_n_heads | 8 |
| | attention_dropout | 0.15 |
| | ffn_d_hidden_multiplier | 1.25 |
| | ffn_dropout | 0.05 |
| | residual_dropout | 0 |
| | learning_rate | $10^{-4}$ |
| | weight_decay | $10^{-5}$ |
| | early_stopping | True |
| Naive Bayes | var_smoothing | $10^{-9}$ |
| $K$-Nearest-Neighbors | n_neighbors | 5 |
| | leaf_size | 30 |
| | p | 2 |
| Logistic Regression | penalty | l2 |
| | tol | $10^{-4}$ |
| | C | 1 |
| | solver | lbfgs |
| MLP | d_hidden | 100 |
| | n_hidden | 1 |
| | learning_rate | $10^{-4}$ |
| | early_stopping | True |
| ResNet | n_blocks | 4 |
| | d_block | 128 |
| | d_hidden_multiplier | 1.25 |
| | dropout | 0.2 |
| | learning_rate | 0.0001 |
| | weight_decay | 0.00001 |
| | early_stopping | True |
| | patience | 16 |

does not have a "centroid", we manually calculate a euclidean centroid for each cluster by using `sklearn.neighbors.NearestCentroid` to compute the centroid or the closest *real* point.

**KIP** We use the implementation provided by Nguyen et al. (2021) available at `https://github.com/google-research/google-research/tree/master/kip`.

**GM** We use the implementation provided by Zhao et al. (2021) available at `https://github.com/VICO-UoE/DatasetCondensation`.

Table 11 shows the parameters available for each distillation methods. The common parameters are used for every algorithm, with the exceptions marked on the right-most column. The method-specific parameters for KIP and GM are for the original algorithms as proposed in Nguyen et al. (2021); Zhao et al. (2021).

### A.6 DOWNSTREAM CLASSIFIER HYPERPARAMETERS

Table 12 shows the hyperparameters used for each downstream classifier. We use scikit-learn Pedregosa et al. (2011)'s implementation of Naive Bayes, $K$-Nearest-Neighbors, Logistic Regression, and MLP, and Gorishniy et al. (2021)'s implementation of FT-Transformer and ResNet.

Table 13: Average train/test times and test performance comparison for all downstream classifiers.

| Classifier | Train Time | Test Time | Test Perf. |
|---|---|---|---|
| FTTransformer | 281.3431 | 0.17934 | 0.7879 |
| NB | 0.0030 | 0.00232 | 0.6624 |
| KNN | 0.0007 | 0.54309 | 0.7474 |
| LR | 0.4901 | 0.00646 | 0.7709 |
| MLP | 2.4444 | 0.00554 | 0.7826 |
| ResNet | 154.9824 | 0.08508 | 0.7833 |
| XGB | 11.4055 | 0.01439 | 0.8180 |

### A.7 RESNET AND FT-TRANSFORMER PERFORMANCE

We test ResNet and FT-Transformer for 5 datasets. We found that even with early stopping, the two classifiers take significantly longer to train given the same computing resources. On average, we find that ResNet takes around 10 times longer to finish training, while FT-Transformer takes around 28 times when compared to XGBoost. We also find that the performance of resnet and FT-Transformer does not stand out – in fact, the average test performance when trained on the full dataset shows that both ResNet and FTTransformer show a similar performance to MLP, and are outperformed by XGBoost.

### A.8 DETERMINING THE BEST OVERALL PERFORMANCE

We describe the best overall pipeline in section 4 and table 2. Here, we provide a more detailed explanation of how we determined the best overall pipeline. The runs are grouped by their classifier, dataset and distill size $n$. Similar to other parts of analysis, the grouping is done in order to ensure that the comparisons are *fair*. In this instance, we are interested in only the pipeline components that lead to the best classifier performance, regardless of the exact classifier kind. Thus, we group every run by their non-pipeline-specific parameters, which are the classifier, dataset and distill size $n$. In each group, we then count the instances the pipeline places on the top 3 in terms of the regret score and sum up the counts for each pipeline.

Following the previous findings, table 2 shows that $k$-means based methods have the best performance, placing in the top 3 with all SFT encoder variants. Surprisingly, we also find pipelines that use KIP and GM as the 4th and 5th best performers. While we were not able to determine any specific conditions that cause KIP and GM to place on top, this result shows that there are exist some conditions which leads the pipelines using gradient-based methods (KIP, GM) to be the top performer. On the other hand, the consistent rank placement of pipelines that use the autoencoder latent space shows that fine-tuned autoencoders can indeed boost the performance of distillation methods significantly.

## B ADDITIONAL ANALYSIS

### B.1 FULL RESULTS OF DISTILLATION METHODS BY DOWNSTREAM CLASSIFIERS

### B.2 EFFECT OF COLUMN EMBEDDING SCHEME ON DOWNSTREAM PERFORMANCE

While column embeddings are standard for categorical columns – each category is represented with a vector, there are various ways of embedding numerical columns: (i) A numerical feature can be binned, and each bin treated as a category with an embedding $\mathbf{w} \in \mathbb{R}^m$ corresponding to each bin. (ii) With linearly scaled column embeddings, a single column embedding $\mathbf{w} \in \mathbb{R}^m$ is used for each numerical column, and the column embedding for a particular numerical value $v \in \mathbb{R}$ is obtained by scaling $\mathbf{w}$ to $v \cdot \mathbf{w}$. (iii) Piecewise linear encoding or PLE (Gorishniy et al., 2022) also bin the numerical feature but use a more sophisticated way of generating the column embeddings for a given numerical value. We considered binned numerical features in the main paper for a couple of reasons: (a) Binned numerical features naturally handle missing values (quite prevalent in tabular data) by maintaining a "missing" bin instead of relying on a heuristic intermediate imputation step; sometimes, the fact that a value is missing is in itself a signal, and heuristic imputation schemes often lose this information. (b) The binned features can be used for all architectures we consider here –

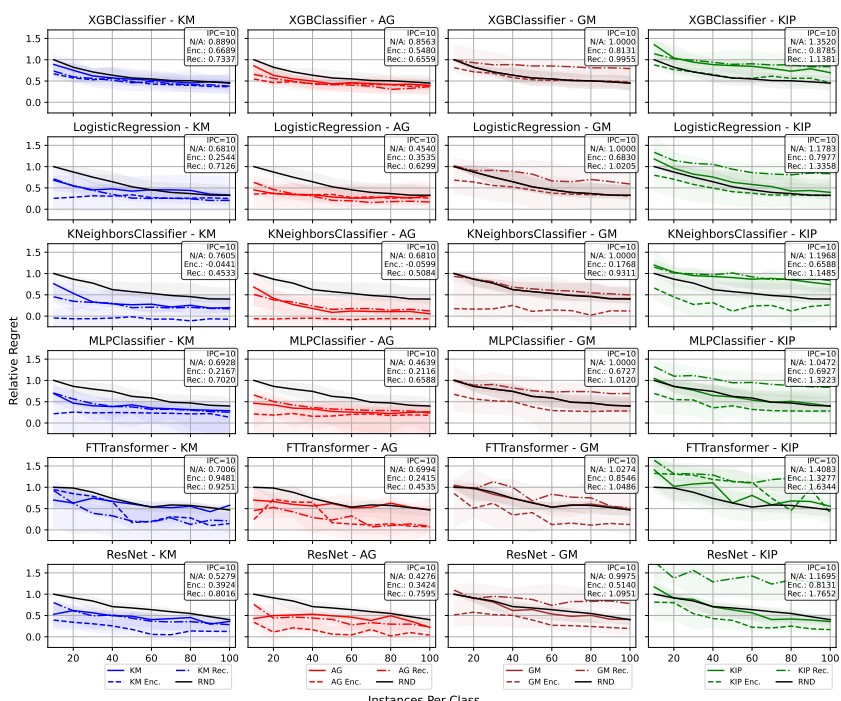

Figure 12: Full results of distillation methods by downstream classifiers.

Table 14: A comparison of relative regret scores of distillation pipelines that use the encoded space of autoencoders trained with different column embeddings, tested on 5 datasets (Adult, Amazon Employee Access, Credit, House, Phishing Websites). The center value shows the median relative regret, and smaller values on each side refers to the first and third quantile, respectively. In general, PLE embeddings show the strongest performance. However, it is worth noting that PLE embeddings are not applicable to GNN encoders, and that binary embeddings also show superior performance to scaled embeddings.

| Col. Emb. | KM | AG | GM | KIP |
|---|---|---|---|---|
| Binary | $_{0.1082}$ 0.5645 $_{0.7886}$ | $_{0.0976}$ 0.4633 $_{0.7181}$ | $_{0.5504}$ 0.9038 $_{1.0063}$ | $_{0.6551}$ 0.9254 $_{1.1918}$ |
| Scaled | $_{0.7214}$ 0.8613 $_{1.0671}$ | $_{0.4908}$ 0.6939 $_{1.0249}$ | $_{1.0092}$ 1.4412 $_{1.8658}$ | $_{1.3304}$ 1.6137 $_{2.2985}$ |
| PLE | $_{-0.2428}$ 0.1976 $_{0.9305}$ | $_{-0.2698}$ 0.2173 $_{0.6752}$ | $_{-0.0865}$ 0.2747 $_{1.0524}$ | $_{-0.0263}$ 0.7398 $_{1.3923}$ |

FFN, Transformer, and GNN – and using a common embedding scheme allows us to ablate the effect of the different architectures. The other numerical embedding schemes do not apply to GNNs.

To understand the effect of different kinds of column embeddings schemes, we conduct a smaller scale experiment on 5 datasets. Specifically, we compare scaled embeddings as seen in Gorishniy et al. (2021), piecewise linear encoding (PLE) as seen in Gorishniy et al. (2022), against using binary column embeddings where continuous features are binarized by binning, and examine the downstream performance of distillation pipelines that use the latent space of the autoencoders trained with the corresponding column embedding scheme. Table 14 shows that using the **both binary column embeddings and PLE consistently leads to lower regret scores compared to scaled column embeddings**. While PLE embeddings show the strongest performance, they are not applicable to the GNN autoencoder architecture. Thus, we conduct most of our experiments using binary column embeddings for a fair comparison across different autoencoder architectures for a fair comparison.

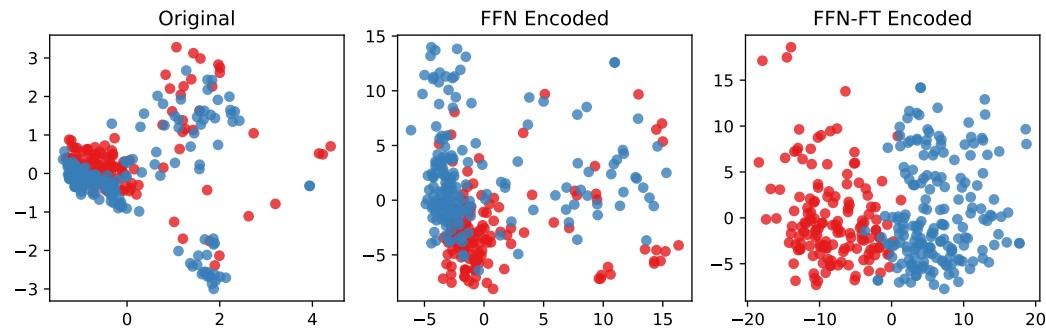

Figure 13: PCA visualization of Phishing Websites dataset.

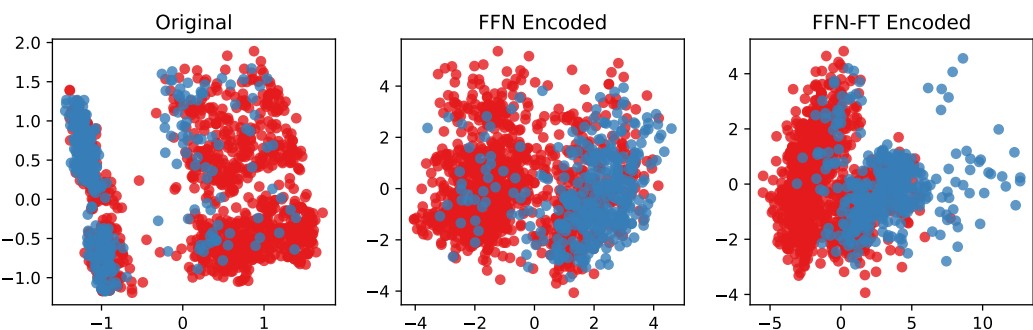

Figure 14: PCA visualization of the Adult dataset.

## B.3 EFFECT OF SUPERVISED FINE-TUNING.

Figures 13 and 14 show the PCA visualizations the adult and tencent CTR datasets in the original, FFN-encoded, FFN-SFT encoded representations. Both figures show that while the distribution inside the vanilla FFN's latent space does not look significantly different from the original space, adding supervised fine-tuning leads to a clearer separation between different classes.

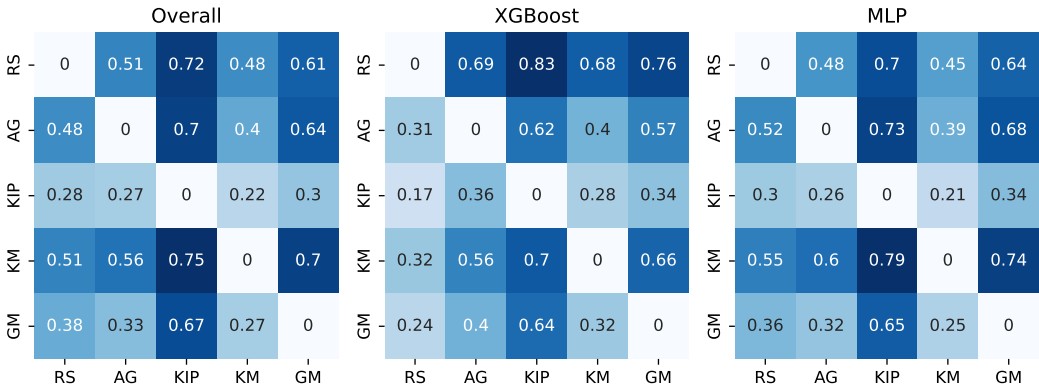

Figure 15: *Pairwise comparision of distillation methods.* The relative performances of distillation methods under otherwise equal sttings. Rows denote *win* ratio, columns denote *loss* ratio.

### B.4 Pairwise Comparision of Distillation Methods.

In addition, we compare the downstream classifier performance with every pair of pipelines that use different distillation methods under otherwise equal settings. The left table of fig. 15 reveals that KIP had the highest tendency to underperform other distillation methods, while *k*-means had the highest tendency to outperform other distillation methods. This is consistent with our previous findings, where *k*-means outranked other distillation methods most frequently. In order to gain further insights behind the performance lag of graident-based distillation methods, we conduct a pairwise comparison of the distillation methods for different classifiers as well. The center and right tables of fig. 15 shows the pairwise comparison of distillation methods for XGBoost and MLP as downstream models. This suggests that gradient-based methods' underperformance is not solely due to its kernel, but that tabular data itself may pose a unique challenge in distillation that is not seen in image data. It is also worth noting that while the clustering-based approaches had the best overall rank, random sampling proved to be a strong baseline with a near 50% win ratio against them.

## C Documentation of TDBench

The information in this section is also available in a markdown format in the `README.md` file of the supplementary material.

### C.1 Reproducing Results

Every plot and table in the main paper can be reconstructed using the following scripts:

- `Q0_experiment_scale.py`
- `Q1_1_col_embeds.py`
- `Q1_encoding.py`
- `Q2_distill_methods.py`
- `Q3_autoencoders.py`
- `Q4_1_runtime.py`
- `Q4_2_get_hpo_dirs.py`
- `Q4_2_hpo.py`
- `Q4_combinations.py`
- `Q5_class_imbal.py`

The scripts are organized in order of the question addressed in section 4 and will be populated in `iclr-figures` directory. These can be simply ran by calling `python SCRIPT_NAME`.

The following files are included in the supplementary material and contain all the necessary information for the scripts:

- `dataset_stats.csv`
- `enc_stats.csv`
- `*data_mode_switch_results.csv`
- `hpo-measure/`
- `*mixed_tf_results.csv`
- `*ple_tf_results.csv`

The files marked with an asterisk (*) are not included in the repository, but can be downloaded from this url: https://drive.google.com/drive/folders/1tJ5e1iCvaz-UbxEgpmuCPj-58crgYRJW?usp=share_link

### C.2 Description of the Workflow

The `## Running the Code` section of `README.md` file discusses the actual commands and available options for running each stage in detail.

The procedure is as follows:

- Train the autoencoder with the desired configuration.
- (*Optional*) Fine-tune the autoencoder with a classifier head.

• Run distillation methods against specified downstream classifiers.

## C.3 CONSTRUCTING A NEW PIPELINE

**Changing default parameters**   The configurations for this project are managed by hydra and can be modified by adding new files/directories under the 'config' directory.

**Adding new datasets**   Adding new datasets is as simple as adding a new `config/data/datasets/DATASET_NAME.yaml` file. Currently, only openml datasets are supported.

| Field | Type |
|---|---|
| dataset_name | string |
| download_url | string |
| label | string |
| n_classes | int |
| source_type | string |

Table 15: Configuration details for datasets

The following flags must be specified for the dataset to be correctly loaded as seen in table 15.

| Field | Type |
|---|---|
| parse_mode | string |
| scale_mode | string |
| bin_strat | string |
| n_bins | int |

Table 16: Configuration details for data preprocessing

**Adding new preprocessing methods**   The preprocessing is handled by the `TabularDataModule` object that lives in `tabdd/data/tabulardatamodule.py`. The preprocessing strategies are identified by a string, and can be configured under `config/data/mode`. The fields seen in table 16 must be specified for the preprocessing to work correctly. One can additionally define any type of `scale_mode` or `bin_strat`, which will be consumed by the `TabularDataModule`.

This object is configured with `DatasetConfig` and `DataModeConfig`. The `DatasetConfig` is the configuration for the dataset, and the `DataModeConfig` is the configuration for the preprocessing method.

It's `TabularDataModule.prepare_data` is the method that will parse the data accordingly and save to cache. One can add arbitrary preprocessing methods in this file by adding new flags to `DataModeConfig` and handling it inside the `prepare_data` method.

| Field | Type |
|---|---|
| is_random | string |
| is_cluster | string |
| can_use_encoder | string |
| args | int |

Table 17: Configuration details for distillation methods

**Adding new distillation methods**   The distillation methods are identified by a string, which should have a configuration with the same name under `config/distill/methods`. Once can characterize the method the following fields seen in table 17.

- `is_random`: Whether there is randomness in the method. If true, the pipeline will be ran multiple times.
- `is_cluster`: Whether the method is a clustering method. If true, an option that uses the nearest-to-center method will be included.
- `can_use_encoder`: Whether the method can be applied in the latent space.
- `args`: any additional arguments to the actual function.

Once the configuration is created, it will be consumed by `load_distilled_data` method of `tabdd/distill/load_distilled_data.py.` This method can then be modified to include the new distillation method.

**Adding new encoders**  All encoders used in the benchmark are subclasses `BaseEncoder` from `tabdd/models/encoder/base_encoder.py`. A simple example of how to implement can be seen in `tabdd/models/encoder/mlp_autoencoder.py`. The module needs to encoder the following methods: `__init__()`, `encode`, `decode` and `forward`.

The autoencoders are specified by the configuration files in `config/encoder/models/`. The class of the encoder is specified by `cls`, and the hyperparameters are specified by `tune_params`.

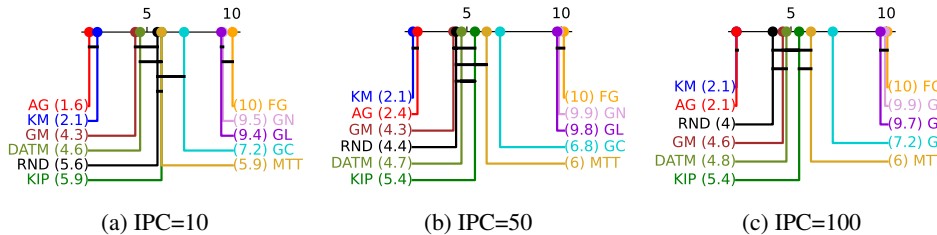

| (a) IPC=10 | (b) IPC=50 | (c) IPC=100 |

Figure 16: Critical difference plot comparing ranks of distillation methods across datasets per IPC value when applied with TF-SFT encoder for XGBoost classifier with additional baselines The x-axis denotes the average rank, and a black horizontal line connects groups of methods that are *not significantly different* in the rank distribution. $k$-means and agglomerative are indistinguishable from each other in IPC $\in \{10, 50\}$, but $k$-means gains an edge in IPC=100. (FG: Forgetting, GN: GraNd, GL: Glister, GC: Graph Cut)

Table 18: Relative regret of pipelines that use different combinations of distill methods and encoders at IPC=10, aggregated over classifiers. The best value for each column is marked with **bold**, and the second best is marked with underline. (FG: Forgetting, GN: GraNd, GL: Glister, GC: Graph Cut)

| Distill Method | Regret | | | | | |
| | Min | Q1 | Mean | Median | Q3 | Max |
| --- | --- | --- | --- | --- | --- | --- |
| Random Sample | 1.0000 | 1.0000 | 1.0000 | 1.0000 | 1.0000 | 1.0000 |
| KM | 0.0597 | 0.5256 | 0.6682 | 0.6654 | **0.8186** | 1.1094 |
| AG | **0.0000** | **0.5177** | **0.6301** | **0.6036** | 0.8914 | **0.9965** |
| KIP | 0.6728 | 0.8483 | 1.1109 | 1.0544 | 1.2523 | 2.2713 |
| GM | 0.4175 | 0.7707 | 0.9858 | 0.9377 | 1.1461 | 1.7292 |
| FG | 0.8705 | 1.1400 | 2.3837 | 1.4465 | 1.8731 | 16.0146 |
| GN | 0.7748 | 1.1498 | 2.0530 | 1.3704 | 2.2624 | 10.6670 |
| GL | 0.8376 | 1.1000 | 2.0907 | 1.3146 | 1.6823 | 14.1625 |
| GC | 0.6361 | 0.9077 | 1.5084 | 1.1031 | 1.5998 | 6.8392 |
| MTT | 0.4175 | 0.7707 | 1.0340 | 0.9699 | 1.2176 | 2.3026 |
| DATM | 0.4175 | 0.7707 | 1.0340 | 0.9699 | 1.2176 | 2.3026 |

# D  ADDITIONAL ANALYSIS

## D.1  ADDITIONAL DISTILLATION METHODS

We conduct a further comparison of more recent distillation methods against the methods compared in section 4 to verify whether these methods will show superior performance. Specifically, we incorporate four representative NN-based coreset selection methods examined in Deepcore (Guo et al., 2022) – Forgetting (Toneva et al., 2018), GraNd (Paul et al., 2021), Glister (Killamsetty et al., 2021), Graph Cut (Iyer & Bilmes, 2013)) and MTT (Cazenavette et al., 2022) and DATM (Guo et al., 2023). The results are presented in Table 18 and Figure 16. Consistent to our findings in section 4, we find that more recent distillation methods that rely on NNs do not fair well on non-differentiable downstream classifier (XGBoost), and that clustering methods still show dominance. It is also interesting to note that GM shows superior performance to MTT and DATM, suggesting that the latter two methods may actually be overfitting to the teacher network's architecture.

## D.2  DATASET FEATURE CORRELATION

We further investigate the presevation of feature correlation in the distilled data. Figure 17 shows the change in feature correlation in the original, randomly sampled and distilled with $k$-means in the latent space of TF-SFT in 3 datasets – Credit, Magic Telescope and Tencent CTR.

## D.3  RELATION TO PREVIOUS WORK

Kang et al. (2024) presented a preliminary abstract on work that explores data distillation for tabular data. The authors utilize an MLP and GNN based autoencoder networks to transform the data

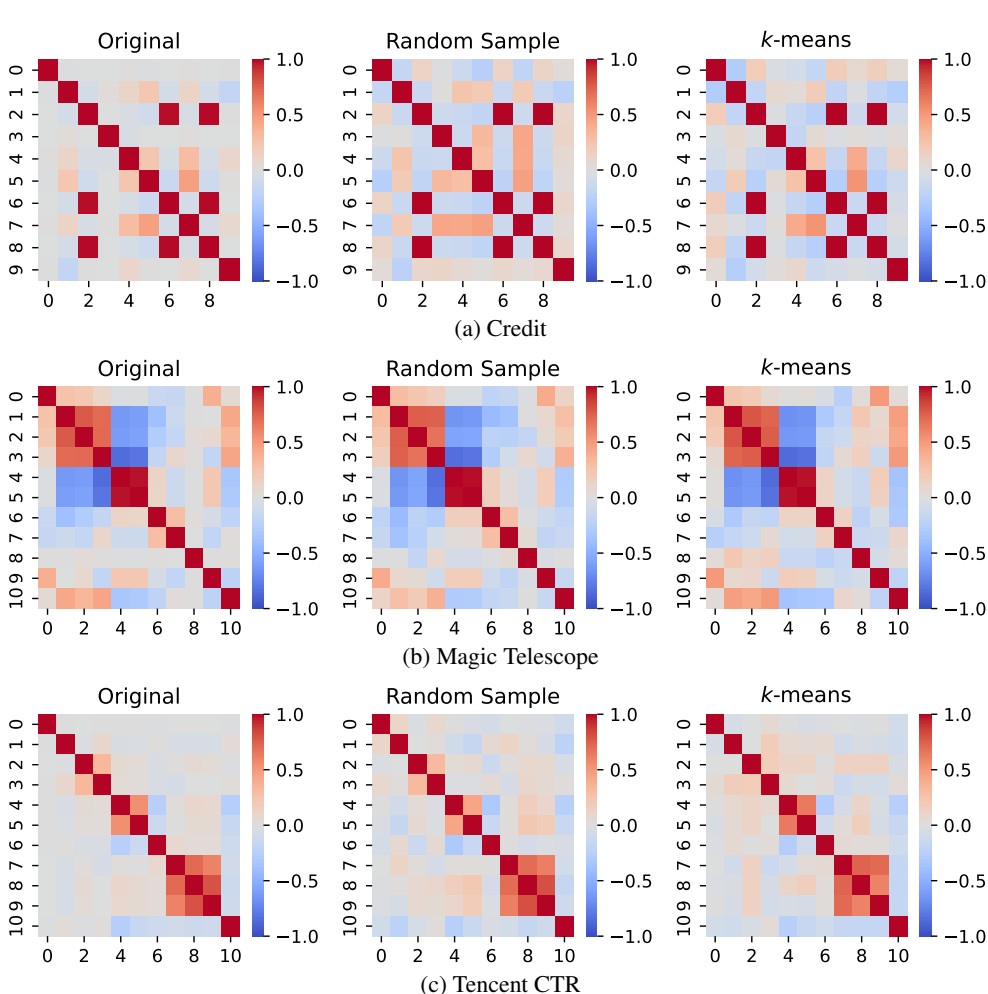

Figure 17: A side-by-side comparison of correlation of numerical features in the training data before distillation, after random sampling@IPC=100, and after distillation@ICP=100. While some weaker correlations are not entirely accurately portrayed, the distilled data preserves the stronger correlations remarkably well.

to before distilling and show that simple clusetering-based methods can outperform competetive distillation algorithms proposed in computer vision (KIP (Nguyen et al., 2021))

Building upon this work, our work provides a comprehensive analysis of distillation methods on tabular data, and provides a detailed comparison of distillation methods across a wide range of datasets and classifiers. We also provide a detailed analysis of the effect of IPC on the performance of distillation methods, and provide insights into the effect of distillation on the feature correlation of the data.

Below, we provide a detailed comparison of our work with the preliminary abstract presented by Kang et al. (2024):

- We conduct a comprehensive comparison of different binning methods and their effect on downstream performance.
- We test with a transformer-based autoencoder, and show that it outperforms MLP and GNN based autoencoders.
- We additionally consider gradient matching Zhao et al. (2021) as an additional baseline to represent the gradient-based family of distillation methods Cazenavette et al. (2022); Zhao & Bilen (2023); Guo et al. (2023)
- We provide a complete python package, TDBench, that can be used and extended by anyone in the community.
- We explore a realistic use case for data distillation in the context of HPO and show the trade-offs in utility and cost saving.
- We introduce a relative regret metric to compare the performance of different distillation methods across datasets and classifiers.

### D.4 RAW BALANCED ACCURACY SCORE

Below is a comparison of the raw balanced accuracy of each distillation pipelines averaged over random iterations. Table 19 shows a comparison of all 10 distillation methods that were ran with TF-SFT encoder and tested on XGB downstream classifier, and table 20 shows the performance of the baseline methods when applied without the encoders. Tables 21 and 22 show the same comparison that with and without TF-SFT encoder for the 4 baselines methods ($k$-means, aggloermative, KIP, GM) on KNN classifier, and tables 23 and 24 show the same for MLP classifier.

The last two rows of the tables each denote the number of instances that the pipeline ranked at the top, and the number of times it outperformed random sampling. The results show that random sampling is not a trivial baseline for many methods, and that both clustering methods, AG and KM, show the strongest performance. We also see that adding the encoder to the pipeline significantly increases the downstream performer of all 3 representative models.

| Dataset | AG | DATM | FG | GM | GL | GN | GC | KIP | KM | MTT |
|---|---|---|---|---|---|---|---|---|---|---|
| AD | 0.7409±0.0128 | 0.7304±0.0316 | 0.6779±0.0458 | 0.7304±0.0316 | 0.5241±0.0000 | 0.5241±0.0000 | 0.6491±0.0545 | 0.7174±0.0517 | **0.7425±0.0220** | 0.7304±0.0316 |
| AE | 0.5072±0.0017 | 0.5233±0.0250 | 0.4960±0.0355 | 0.5253±0.0308 | 0.4950±0.0086 | 0.5149±0.0348 | **0.5418±0.0119** | 0.5107±0.0228 | 0.5292±0.0210 | 0.5233±0.0250 |
| BM | 0.7519±0.0258 | **0.7619±0.0124** | 0.6247±0.0878 | **0.7619±0.0124** | 0.3536±0.0325 | 0.5063±0.0007 | 0.7204±0.0387 | 0.7113±0.0425 | 0.7418±0.0110 | **0.7619±0.0124** |
| CR | 0.5215±0.0079 | 0.5463±0.0119 | 0.4606±0.0078 | 0.5463±0.0119 | 0.5051±0.0097 | 0.4990±0.0057 | 0.4872±0.0483 | **0.5504±0.0089** | 0.5400±0.0137 | 0.5463±0.0119 |
| CD | 0.5932±0.0348 | 0.5900±0.0430 | 0.3826±0.0299 | 0.5900±0.0430 | 0.5334±0.0141 | 0.5088±0.0556 | 0.5703±0.0243 | 0.5761±0.0484 | **0.6083±0.0173** | 0.5900±0.0430 |
| DB | 0.5147±0.0023 | **0.5288±0.0366** | 0.4327±0.0089 | **0.5288±0.0366** | 0.4909±0.0226 | 0.4909±0.0226 | 0.5256±0.0454 | 0.5185±0.0485 | 0.5275±0.0205 | **0.5288±0.0366** |
| EL | **0.6305±0.0015** | 0.5464±0.0279 | 0.5155±0.0392 | 0.5464±0.0279 | 0.5049±0.0015 | 0.5044±0.0017 | 0.4855±0.0377 | 0.5671±0.0432 | 0.6148±0.0160 | 0.5464±0.0279 |
| EV | 0.6432±0.0100 | 0.6094±0.0385 | 0.5640±0.0621 | 0.6094±0.0385 | 0.5544±0.0258 | 0.5544±0.0258 | 0.5664±0.0211 | 0.6139±0.0535 | **0.6504±0.0328** | 0.6094±0.0385 |
| HG | **0.5894±0.0069** | 0.5270±0.0213 | 0.4666±0.0281 | 0.5270±0.0213 | 0.4992±0.0145 | 0.4728±0.0333 | 0.5064±0.0176 | 0.5174±0.0415 | 0.5818±0.0090 | 0.5270±0.0213 |
| HE | **0.6510±0.0222** | 0.6239±0.0276 | 0.3396±0.0264 | 0.6239±0.0276 | 0.5216±0.0358 | 0.4571±0.1091 | 0.6108±0.0226 | 0.6115±0.0590 | 0.6377±0.0158 | 0.6239±0.0276 |
| HS | 0.7387±0.0086 | 0.6865±0.0168 | 0.3574±0.0702 | 0.6865±0.0168 | 0.5718±0.0408 | 0.5718±0.0408 | 0.5814±0.0661 | 0.6725±0.0228 | **0.7417±0.0136** | 0.6865±0.0168 |
| JN | 0.6818±0.0176 | 0.6609±0.0371 | 0.3574±0.0702 | 0.6609±0.0371 | 0.4908±0.0569 | 0.4213±0.1246 | 0.6051±0.0691 | 0.6773±0.0164 | **0.6883±0.0232** | 0.6609±0.0371 |
| LA | **0.8821±0.0363** | 0.8800±0.0199 | 0.8246±0.0606 | 0.8800±0.0199 | 0.5769±0.1078 | 0.5769±0.1078 | 0.6681±0.0566 | 0.8692±0.0485 | 0.8504±0.0403 | 0.8800±0.0199 |
| MT | 0.9303±0.0194 | 0.9423±0.0063 | 0.5985±0.4075 | 0.9423±0.0063 | 0.6450±0.2563 | 0.7326±0.0261 | 0.8285±0.0853 | **0.9620±0.0108** | 0.9358±0.0135 | 0.9423±0.0063 |
| MA | 0.5457±0.0007 | 0.5264±0.0153 | 0.5086±0.0347 | **0.5524±0.0273** | 0.5098±0.0150 | 0.5098±0.0150 | 0.5483±0.0147 | 0.5354±0.0073 | 0.5396±0.0163 | 0.5264±0.0153 |
| MB | 0.6482±0.0196 | 0.6649±0.0406 | 0.4087±0.1227 | 0.6649±0.0406 | 0.5201±0.0278 | 0.5172±0.0492 | 0.5955±0.0767 | 0.6665±0.311 | **0.6793±0.0147** | 0.6649±0.0406 |
| NU | **0.5037±0.0058** | 0.5012±0.0040 | 0.5009±0.0061 | 0.5012±0.0040 | 0.4985±0.0042 | 0.4978±0.0049 | 0.5032±0.0039 | 0.5022±0.0052 | 0.5025±0.0042 | 0.5012±0.0040 |
| NS | 0.8964±0.0237 | 0.9126±0.0412 | 0.8325±0.0980 | 0.9126±0.0412 | 0.6211±0.0601 | 0.6211±0.0601 | 0.8335±0.0142 | **0.9272±0.0161** | 0.9023±0.0225 | 0.9126±0.0412 |
| PW | 0.7976±0.0044 | **0.8198±0.0131** | 0.5660±0.1908 | 0.7698±0.0321 | 0.6195±0.0669 | 0.6311±0.0485 | 0.6609±0.1233 | 0.6843±0.0690 | 0.7812±0.0388 | **0.8198±0.0131** |
| PL | 0.7954±0.0533 | 0.7718±0.0735 | 0.6692±0.1175 | 0.7718±0.0735 | 0.6462±0.0636 | 0.6462±0.0636 | 0.6713±0.0753 | 0.7373±0.0813 | **0.8128±0.0264** | 0.7718±0.0735 |
| RS | 0.6528±0.0048 | 0.6247±0.0217 | 0.3369±0.0384 | 0.6247±0.0217 | 0.5454±0.0446 | 0.5456±0.0444 | 0.5881±0.0591 | 0.5697±0.0666 | **0.6616±0.0226** | 0.6247±0.0217 |
| TC | 0.5536±0.0037 | 0.5110±0.0251 | 0.5219±0.0236 | 0.5108±0.0153 | 0.5262±0.0116 | 0.5342±0.0155 | 0.5298±0.0082 | 0.5307±0.0278 | **0.5623±0.0153** | 0.5110±0.0251 |
| TD | 0.8543±0.0033 | 0.8568±0.0224 | 0.2951±0.1540 | **0.8670±0.0205** | 0.7513±0.1152 | 0.3936±0.3638 | 0.7600±0.1364 | 0.8055±0.0451 | 0.8563±0.0080 | 0.8568±0.0224 |
| # Best | 5/23 | 3/23 | 0/23 | 4/23 | 0/23 | 0/23 | 1/23 | 3/23 | 9/23 | 3/23 |
| vs RND | 18/23 | 18/23 | 6/23 | 18/23 | 2/23 | 2/23 | 11/23 | 18/23 | 18/23 | 18/23 |

Table 19: Comparison of raw balanced accuracy scores of distillation methods applied with TF-SFT on XGB classifier. Last two rows of the tables each denote the number of instances that the pipeline ranked at the top, and the number of times it outperformed random sampling. Best performance at for each dataset is marked in bold, and second-best performance is marked with underline.

| Dataset | AG | GM | KIP | KM |
|---------|-----|-----|-----|-----|
| AD | 0.6078±0.0354 | **0.7175±0.0423** | 0.5353±0.0562 | 0.5949±0.0644 |
| AE | 0.5252±0.0195 | **0.5304±0.0167** | 0.5060±0.0066 | 0.5007±0.0126 |
| BM | 0.5599±0.0525 | 0.5816±0.0448 | 0.5114±0.0179 | **0.5842±0.0914** |
| CR | **0.5667±0.0491** | 0.5394±0.0334 | 0.5106±0.0383 | 0.5612±0.0578 |
| CD | **0.5887±0.0310** | 0.5607±0.0448 | 0.5309±0.0466 | 0.5526±0.0382 |
| DB | 0.5133±0.0157 | 0.5053±0.0265 | 0.5008±0.0140 | **0.5146±0.0137** |
| EL | **0.5929±0.0782** | 0.5617±0.0545 | 0.5093±0.0290 | 0.5866±0.0741 |
| EV | **0.6120±0.0658** | 0.5968±0.0567 | 0.5730±0.0590 | 0.6009±0.0754 |
| HG | **0.5159±0.0107** | 0.5130±0.0143 | 0.5028±0.0094 | 0.5141±0.0196 |
| HE | 0.5909±0.0372 | **0.5918±0.0378** | 0.5112±0.0396 | 0.5790±0.0608 |
| HS | **0.6770±0.0526** | 0.6257±0.0567 | 0.5288±0.0671 | 0.6484±0.0956 |
| JN | 0.6076±0.0176 | **0.6111±0.0262** | 0.5759±0.0654 | 0.5755±0.0511 |
| LA | 0.8079±0.1752 | 0.8006±0.1236 | 0.7352±0.1598 | **0.8101±0.1533** |
| MT | 0.8217±0.1813 | **0.9581±0.0285** | 0.8029±0.1473 | 0.8082±0.1785 |
| MA | 0.5146±0.0185 | **0.5585±0.0324** | 0.4991±0.0108 | 0.5112±0.0222 |
| MB | **0.6715±0.0942** | 0.6480±0.0710 | 0.5559±0.0732 | 0.6476±0.1124 |
| NU | **0.5047±0.0079** | 0.5005±0.0060 | 0.5004±0.0041 | 0.5022±0.0050 |
| NS | **1.0000±0.0000** | **1.0000±0.0000** | **1.0000±0.0000** | **1.0000±0.0000** |
| PW | 0.7665±0.1429 | **0.8466±0.0613** | 0.6758±0.1216 | 0.7918±0.1242 |
| PL | 0.5966±0.0441 | 0.6813±0.0515 | 0.6045±0.1043 | **0.6834±0.0898** |
| RS | **0.6469±0.0546** | 0.5810±0.0373 | 0.5200±0.0350 | 0.6469±0.0541 |
| TC | **0.5343±0.0295** | 0.5118±0.0357 | 0.5031±0.0228 | 0.5301±0.0245 |
| TD | **0.8162±0.0100** | 0.7790±0.0270 | 0.6355±0.0884 | 0.7736±0.0584 |
| # Best | 12/23 | 8/23 | 1/23 | 5/23 |
| vs RND | 15/23 | 16/23 | 3/23 | 15/23 |

Table 20: Comparison of raw balanced accuracy scores of distillation methods applied in the original space (no encoder) on XGB classifier. Last two rows of the tables each denote the number of instances that the pipeline ranked at the top, and the number of times it outperformed random sampling. Best performance at for each dataset is marked in bold, and second-best performance is marked with underline.

| Dataset | AG | GM | KIP | KM |
|---|---|---|---|---|
| AD | 0.7904±0.0171 | 0.7609±0.0170 | 0.6645±0.0662 | **0.7940±0.0078** |
| AE | **0.5371±0.0022** | 0.5246±0.0244 | 0.5129±0.0130 | 0.5365±0.0192 |
| BM | **0.7898±0.0052** | 0.7546±0.0317 | 0.6997±0.0556 | 0.7897±0.0083 |
| CR | 0.5437±0.0127 | 0.5337±0.0260 | **0.5500±0.0170** | 0.5219±0.0199 |
| CD | 0.6490±0.0302 | 0.6449±0.0471 | 0.5819±0.0483 | **0.6674±0.0112** |
| DB | **0.5607±0.0019** | 0.5054±0.0474 | 0.5408±0.0335 | 0.5565±0.0064 |
| EL | 0.6163±0.0173 | 0.5758±0.0423 | 0.5655±0.0241 | **0.6276±0.0126** |
| EV | **0.7152±0.0017** | 0.6621±0.0319 | 0.6205±0.0448 | 0.7130±0.0193 |
| HG | **0.5796±0.0338** | 0.5239±0.0128 | 0.5205±0.0106 | 0.5792±0.0130 |
| HE | **0.6870±0.0061** | 0.6588±0.0174 | 0.6325±0.0600 | 0.6786±0.0103 |
| HS | **0.7759±0.0119** | 0.7211±0.0279 | 0.6575±0.0831 | 0.7721±0.0128 |
| JN | **0.7383±0.0050** | 0.6972±0.0111 | 0.6795±0.0142 | 0.7308±0.0035 |
| LA | **0.9979±0.0000** | 0.9654±0.0255 | 0.9395±0.0760 | 0.9935±0.0055 |
| MT | **0.9717±0.0002** | 0.9674±0.0040 | 0.9714±0.0065 | 0.9715±0.0026 |
| MA | 0.5570±0.0096 | 0.5587±0.0252 | 0.5063±0.0160 | **0.5683±0.0083** |
| MB | 0.6478±0.0156 | 0.6871±0.0152 | 0.6307±0.0494 | **0.6939±0.0241** |
| NU | 0.4971±0.0083 | 0.4994±0.0060 | 0.4967±0.0058 | **0.5075±0.0020** |
| NS | **0.9944±0.0063** | 0.9573±0.0063 | 0.9716±0.0095 | 0.9941±0.0056 |
| PW | 0.8964±0.0158 | 0.8620±0.0170 | 0.6696±0.0283 | **0.9016±0.0158** |
| PL | 0.7829±0.0206 | 0.7505±0.0500 | 0.6717±0.0584 | **0.8277±0.0313** |
| RS | 0.7154±0.0216 | 0.6357±0.0386 | 0.6679±0.0459 | **0.7208±0.0134** |
| TC | 0.5530±0.0256 | 0.5261±0.0222 | 0.5173±0.0106 | **0.5609±0.0138** |
| TD | 0.9230±0.0012 | 0.9117±0.0167 | 0.8204±0.0502 | **0.9242±0.0052** |
| # Best | 11/23 | 0/23 | 1/23 | 11/23 |
| vs RND | 22/23 | 21/23 | 15/23 | 22/23 |

Table 21: Comparison of raw balanced accuracy scores of distillation methods with TF-SFT and KNN downstream classifier. Best performance at for each dataset is marked in bold, and second-best performance is marked with underline.

| Dataset | AG | GM | KIP | KM |
|---|---|---|---|---|
| AD | **0.7352±0.0212** | 0.7292±0.0136 | 0.5600±0.0599 | 0.7246±0.0309 |
| AE | **0.5309±0.0252** | 0.5204±0.0109 | 0.5131±0.0235 | 0.5143±0.0162 |
| BM | 0.7111±0.0136 | 0.6352±0.0335 | 0.5374±0.0504 | **0.7210±0.0277** |
| CR | 0.5364±0.0178 | 0.5393±0.0256 | 0.5161±0.0210 | **0.5508±0.0180** |
| CD | **0.6082±0.0252** | 0.6005±0.0292 | 0.5525±0.0275 | 0.6005±0.0312 |
| DB | 0.5154±0.0162 | 0.5139±0.0261 | 0.5049±0.0213 | **0.5288±0.0178** |
| EL | **0.6300±0.0314** | 0.5630±0.0346 | 0.5273±0.0463 | 0.6210±0.0331 |
| EV | 0.6840±0.0477 | 0.6380±0.0361 | 0.5907±0.0423 | **0.6949±0.0317** |
| HG | **0.5397±0.0181** | 0.5121±0.0124 | 0.5118±0.0064 | 0.5281±0.0128 |
| HE | 0.6442±0.0253 | 0.5837±0.0354 | 0.5194±0.0449 | **0.6546±0.0183** |
| HS | 0.6954±0.0497 | 0.6340±0.0532 | 0.5274±0.0498 | **0.7115±0.0275** |
| JN | 0.6555±0.0197 | 0.6320±0.0185 | 0.5891±0.0515 | **0.6597±0.0211** |
| LA | **0.8267±0.0424** | 0.7451±0.0585 | 0.8233±0.0684 | 0.8039±0.0672 |
| MT | 0.8070±0.0590 | 0.7332±0.0815 | 0.7098±0.0900 | **0.8236±0.0855** |
| MA | **0.5756±0.0098** | 0.5632±0.0237 | 0.5111±0.0340 | 0.5676±0.0186 |
| MB | 0.6712±0.0703 | 0.6122±0.0577 | 0.5565±0.0573 | **0.6731±0.0627** |
| NU | **0.5065±0.0033** | 0.5019±0.0050 | 0.5005±0.0054 | 0.5035±0.0049 |
| NS | 0.9278±0.0753 | 0.8064±0.0162 | **0.9775±0.0140** | 0.8876±0.0842 |
| PW | **0.8700±0.0175** | 0.8128±0.0311 | 0.6291±0.0598 | 0.8678±0.0240 |
| PL | 0.6327±0.0557 | 0.5675±0.0262 | 0.5634±0.0362 | **0.6554±0.0705** |
| RS | **0.6350±0.0324** | 0.5440±0.0214 | 0.5213±0.0200 | 0.6261±0.0304 |
| TC | 0.5129±0.0285 | 0.5152±0.0240 | 0.4953±0.0155 | **0.5205±0.0195** |
| TD | 0.7632±0.0386 | 0.7125±0.0293 | 0.6139±0.0481 | **0.7814±0.0377** |
| # Best | 10/23 | 0/23 | 1/23 | 12/23 |
| vs RND | 22/23 | 18/23 | 3/23 | 22/23 |

Table 22: Comparison of raw balanced accuracy scores of distillation methods in the original space (no encoder) KNN downstream classifier. Best performance at for each dataset is marked in bold, and second-best performance is marked with underline.

| Dataset | AG | GM | KIP | KM |
|---|---|---|---|---|
| AD | 0.7627±0.0039 | 0.7406±0.0168 | 0.7318±0.0212 | **0.7628±0.0210** |
| AE | 0.5467±0.0250 | 0.5324±0.0090 | 0.5189±0.0063 | **0.5630±0.0212** |
| BM | **0.7894±0.0319** | 0.7685±0.0210 | 0.7632±0.0312 | 0.7887±0.0142 |
| CR | 0.5299±0.0242 | 0.5443±0.0228 | **0.5525±0.0185** | 0.5349±0.0134 |
| CD | 0.6323±0.0845 | 0.6358±0.0411 | 0.6138±0.0261 | **0.6542±0.0414** |
| DB | 0.5364±0.0051 | 0.5211±0.0278 | 0.5348±0.0349 | **0.5405±0.0139** |
| EL | 0.6432±0.0317 | 0.5690±0.0326 | 0.6131±0.0309 | **0.6543±0.0187** |
| EV | **0.7310±0.0019** | 0.6792±0.0421 | 0.6742±0.0347 | 0.7202±0.0350 |
| HG | **0.6058±0.0126** | 0.5302±0.0068 | 0.5477±0.0254 | 0.5997±0.0152 |
| HE | 0.6540±0.0057 | 0.6364±0.0225 | 0.6256±0.0370 | **0.6580±0.0157** |
| HS | **0.7801±0.0029** | 0.7257±0.0336 | 0.7478±0.0149 | 0.7768±0.0141 |
| JN | **0.7192±0.0036** | 0.6911±0.0130 | 0.6952±0.0315 | 0.7153±0.0102 |
| LA | **0.9983±0.0010** | 0.9893±0.0192 | 0.9883±0.0238 | 0.9980±0.0017 |
| MT | 0.9698±0.0055 | 0.9627±0.0056 | 0.9697±0.0050 | **0.9733±0.0032** |
| MA | 0.5694±0.0127 | 0.5571±0.0283 | 0.5160±0.0107 | **0.5878±0.0149** |
| MB | **0.6818±0.0092** | 0.6555±0.0568 | 0.6697±0.0217 | 0.6707±0.0169 |
| NU | 0.4958±0.0047 | 0.5012±0.0057 | 0.4987±0.0060 | **0.5071±0.0045** |
| NS | 0.9749±0.0153 | 0.9731±0.0179 | 0.9838±0.0139 | **0.9842±0.0129** |
| PW | 0.8804±0.0107 | 0.8466±0.0383 | 0.7921±0.0523 | **0.9046±0.0108** |
| PL | **0.9010±0.0198** | 0.8502±0.0175 | 0.8198±0.0426 | 0.9000±0.0059 |
| RS | 0.6842±0.0019 | 0.6210±0.0500 | 0.6627±0.0714 | **0.6877±0.0205** |
| TC | **0.5785±0.0231** | 0.5150±0.0307 | 0.5366±0.0251 | 0.5734±0.0197 |
| TD | 0.9191±0.0104 | 0.9010±0.0260 | 0.8999±0.0213 | **0.9200±0.0037** |
| # Best | 9/23 | 0/23 | 1/23 | 13/23 |
| vs RND | 21/23 | 19/23 | 18/23 | 23/23 |

Table 23: Comparison of raw balanced accuracy scores of distillation methods with TF-SFT and MLP downstream classifier. Best performance at for each dataset is marked in bold, and second-best performance is marked with underline.

| Dataset | AG | GM | KIP | KM |
|---|---|---|---|---|
| AD | 0.7183±0.0392 | **0.7576±0.0148** | 0.6756±0.0604 | 0.7385±0.0276 |
| AE | **0.5743±0.0265** | 0.5444±0.0153 | 0.5267±0.0241 | 0.5618±0.0410 |
| BM | **0.7406±0.0224** | 0.6573±0.0312 | 0.5776±0.0569 | 0.7351±0.0311 |
| CR | 0.5607±0.0216 | 0.5388±0.0272 | 0.5037±0.0440 | **0.5618±0.0277** |
| CD | **0.6146±0.0276** | 0.5920±0.0524 | 0.5706±0.0564 | 0.6040±0.0332 |
| DB | 0.5168±0.0171 | 0.5203±0.0207 | 0.5052±0.0281 | **0.5329±0.0203** |
| EL | **0.6713±0.0315** | 0.5904±0.0363 | 0.5568±0.0787 | 0.6573±0.0347 |
| EV | 0.6828±0.0270 | 0.6570±0.0289 | 0.6380±0.0651 | **0.6900±0.0255** |
| HG | **0.5463±0.0218** | 0.5184±0.0120 | 0.5190±0.0163 | 0.5423±0.0221 |
| HE | 0.6221±0.0210 | 0.6213±0.0397 | 0.5369±0.0574 | **0.6309±0.0227** |
| HS | **0.7514±0.0159** | 0.6746±0.0321 | 0.5970±0.0850 | 0.7397±0.0396 |
| JN | **0.6352±0.0188** | 0.6328±0.0214 | 0.6209±0.0370 | 0.6339±0.0137 |
| LA | 0.8530±0.0389 | 0.7621±0.0419 | **0.8924±0.1000** | 0.7970±0.0443 |
| MT | **0.9008±0.0332** | 0.8068±0.0741 | 0.8904±0.0269 | 0.8839±0.0584 |
| MA | **0.5710±0.0124** | 0.5591±0.0198 | 0.5294±0.0387 | 0.5640±0.0151 |
| MB | **0.7411±0.0577** | 0.6768±0.0636 | 0.5995±0.0945 | 0.7248±0.0656 |
| NU | **0.5076±0.0025** | 0.5009±0.0057 | 0.5004±0.0028 | 0.5063±0.0059 |
| NS | 0.9799±0.0208 | 0.8159±0.0102 | **0.9967±0.0038** | 0.9006±0.0746 |
| PW | **0.9018±0.0178** | 0.8248±0.0253 | 0.8084±0.0546 | 0.8775±0.0323 |
| PL | 0.7934±0.0946 | 0.6883±0.0491 | 0.7319±0.0239 | **0.7961±0.0719** |
| RS | 0.6304±0.0134 | 0.5567±0.0191 | 0.5386±0.0350 | **0.6305±0.0228** |
| TC | **0.5404±0.0253** | 0.5177±0.0141 | 0.5016±0.0358 | 0.5154±0.0328 |
| TD | **0.7924±0.0154** | 0.7164±0.0507 | 0.6930±0.0413 | 0.7751±0.0265 |
| # Best | 14/23 | 1/23 | 2/23 | 6/23 |
| vs RND | 22/23 | 19/23 | 6/23 | 21/23 |

Table 24: Comparison of raw balanced accuracy scores of distillation methods with in the original space (no encoder) MLP downstream classifier. Best performance at for each dataset is marked in bold, and second-best performance is marked with underline.

