# OpenReview forum: "On Learning Representations for Tabular Dataset Distillation"
_ICLR.cc/2025/Conference — Submitted to ICLR 2025_

### Official Review · Reviewer_fzEk · 2024-10-27

**Soundness:** 2
**Presentation:** 2
**Contribution:** 2
**Rating:** 5
**Confidence:** 3

**Summary:**

The paper introduces TDColER, a novel framework designed for distilling tabular data. It addresses the challenges of feature heterogeneity and non-differentiable models commonly used with tabular data. The authors propose using column embeddings and representation learning to enhance the quality of distilled data. The paper also introduces TDBench, a benchmark for evaluating distillation methods on tabular data. Through extensive experiments, the authors demonstrate TDColER's ability to significantly improve the quality of distilled datasets across various models and datasets.

**Strengths:**

1. The paper introduces TDColER, a tabular data distillation method that effectively addresses key challenges in this field, such as inherent feature heterogeneity and the common use of non-differentiable learning models.
2. The authors construct TDBench, a tabular data distillation benchmark that includes a variety of distilled datasets and models, providing valuable resources for the research community.

**Weaknesses:**

1. The main motivation for conducting tabular dataset distillation is not adequately articulated. The authors should clarify the importance of this area and the specific challenges they aim to address with their approach. A clear rationale for the need for tabular dataset distillation would significantly strengthen the paper's foundation.
2. The paper occasionally lacks clarity in explaining key terms and concepts. For instance, on page 3, the authors mention "feature heterogeneity" and "non-differentiable learning models" without providing clear definitions. This lack of clarity may hinder the readability and accessibility of the research for readers unfamiliar with the nuances of tabular data processing.
3. The experimental section does not sufficiently demonstrate the superiority of TDColER compared to existing SOTA methods such as RDED and DATM.
4. The experimental section fails to provide a comparative analysis that highlights TDColER's efficiency and scalability relative to existing methods, such as the algorithm's complexity, resource consumption, and performance on larger dataset scales.
5. The paper does not establish a theoretical foundation to explain the effectiveness of TDColER, particularly in terms of how it addresses feature heterogeneity and non-differentiable models.

**Questions:**

See weakness.

---

> ### Author Response · Authors · 2024-11-21
>
> We thank you for your detailed review and insightful comments. We have addressed each of your concerns pointed out in the weakness below. It would be great to hear your feedback on our responses.
>
> **Weakness 1: Unclear motivation for tabular data distillation**
>
> Thank you for pointing this out. We apologize for any lack of clarity in our motivation behind this work. We believe that data distillation for tabular data is an important aspect that has many implications, such as storage/operational cost saving, as well as a privacy mechanism. [1] also points out the need for more research into tabular data, a modality that has been overlooked in comparison to images and text in recent days. Thus, the main rationale behind our work is the following: "Can we apply data distillation techniques developed on image datasets directly to tabular data? If not, what adjustments are needed/helpful?". The  main challenges in applying these methods are as mentioned in Section 2: Feature heterogeneity and model agnosticity, where the model may not be a differentiable (NN-based) model. We also further clarify our usage of these two terms in the next response. Our goal is the examine whether these challenges are indeed valid when applying distillation methods directly on tabular data -- and our experiments show that this is indeed the case, where the traditional SotA model (XGBoost) does not gain as much from NN and architecture-specific distillation algorithms. Our results also show that the feature heterogeneity is indeed another challenge for all distillation algorithms, demonstrating a significant improvement in downstream performance when applying TDColER (latent space projection) in combination with the distillation algorithms.
>
> Furthermore, we show that there is indeed a merit in applying data distillation on tabular data in our HPO use case, which showed that conducting HPO with the distilled data on averages uses only 21.84\% of the runtime compared to using the full data, while yielding up to 98.37\% of the performance. This can be a critical benefit in time-sensitive scenarios where some trade-off between computation time and performance is acceptable. We hope that this clarifies our motivation and contributions.
>
> **Weakness 2: Lacking clarity in key terms and concepts**
>
> Thank you for pointing this out. We apologize for any confusion in the terms we used in this work. To further clarify what we meant, below are a more elaborate description of the two issues that are particular to tabular data:
>
> *Feature heterogeneity*: In image data, each feature corresponds to a pixel. In language, token. But in tabular data, it is not as clear as some features may be categorical/ numerical or even binary, and have missing values. Thus, a *feature* in tabular datasets can mean many different things in terms of its semantics, data type, and also its magnitude. Past work has identified that this is one of the key factors of tabular data that causes NNs to struggle when compared to gradient boosting models [2].
>
> *Non-differentiable model*: We realize that we never formally defined this term. When we say non-differentiable learning model, we are referring to any ML model which cannot be used to produce gradients of the prediction output with respect to the input (i.e. cannot be learned through back-propagation). We mainly use this term to differentiate between NN-based models and those that are not NN-based, which essentially refers to XGBoost, K-Nearest Neighbors and Gaussian Naive Bayes in our work. We apologize any confusion in using this term and will update the manuscript with a more clear definition.
>
> **Weakness 3: Is distillation of tabular data alone worth it?**
>
> Thank you for suggesting these works. While [3] is not directly applicable to tabular data due to their usage of image *patches*, which is not a concept that translates natively to tabular data, we have incorporated [4], and its predecessor [5] in to our additional analysis. Please find the detailed report that was added to Appendix D.1.
>
> In addition, we would like to note TDColER itself is not a distillation algorithm -- rather, it is a wrapper on top of an arbitrary distillation algorithm to improve its application on tabular data. This was the reason we only considered off-the-shelf methods and test the difference in their effectiveness with and without TDColER. Our experiment results also show that distillation methods developed for vision data do not apply very well on tabular data, especially when the downstream classifier is not an NN model. We would also like to highlight that another main contribution of this work -- the TDBench python package -- allows for a quick prototyping and implemention of new distillation methods and can be easily extended by the community in the future as well.

---

> ### Author Response · Authors · 2024-11-21
>
> **Weakness 4: Comparative analysis of TDColER against existing methods**
>
> We apologize for the lack of clarity and thank you for raising this point. Our proposed method, TDColER, by itself is not a distillation algorithm, so the main scalability of the distillation process depends on the core distillation method being applied. Previous work that compared distillation methods [6,7] place emphasis on the downstream model performance on the distilled data, which is why we also focused on downstream predictive performance. We would also like to note that the cost of distillation is a one-time occurrence, as the distilled data can be re-used for multiple downstream tasks.
>
> The additional resource constraint TDColER adds comes from the encoding and decoding process. As detailed in Appendix A.4.3, all three of our encoder architectures' parameter count linearly scales with $D$, the dimension (number of columns) of the original dataset. Similarly, the encoding and decoding steps also scale linearly with respect to $D$, as the parameter size of all autoencoders are used once during forward pass. We also empirically examine the difference between the autoencoder architectures after HPO on the datasets in Figure 4.
>
> **Weakness 5: Theoretical foundations of TDColER**
>
> We thank you for this suggestion. This is indeed an important aspect that we have not considered in this work. However, this is a complex topic that requires a thorough analysis and our work focuses on the downstream models' performance. We believe that the effectiveness of TDColER can be attributed to the ability of the column embeddings to capture the relationships between different columns, which is crucial for tabular data analysis and the ability to adapt to any distillation methods that are not tied to a particular model architecture.
>
> Thus, some examples of existing lines of work that can be incorporated to theoretically explaining the effectiveness of TDColER in future works are the following: feature heterogeneity and distribution shift between the generating and downstream models. Feature heterogeneity is a well-known challenge in machine learning [8]. In tabular data, this is particularly challenging as the features can be of different types (categorical, numerical, etc.) and have different scales. In addition, data distillation works proposed for vision datasets make the assumption of NN-based downstream classifiers, which is not applicable in tabular data. Thus, we believe that existing literature on distribution shift may be a good starting point to understand the shift that occurs between the models that are used to generate the data and the models that are trained on the generated data.
>
> **References**
>
> [1]: "Why Tabular Foundation Models Should Be a Research Priority." ICML 2024 Position Paper
>
> [2]: "Why do tree-based models still outperform deep learning on typical tabular data?" NeurIPS 2022
>
> [3]: "On the Diversity and Realism of Distilled Dataset: An Efficient Dataset Distillation Paradigm." CVPR 2024.
>
> [4]: "Towards Lossless Dataset Distillation via Difficulty-Aligned Trajectory Matching." ICLR 2024.
>
> [5]: "Dataset Distillation by Matching Training Trajectories." CVPR 2022.
>
> [6]: "DC-BENCH: Dataset Condensation Benchmark." NeurIPS Datasets and Benchmark Track, 2022.
>
> [7]: "New Properties of the Data Distillation Method When Working With Tabular Data." AIST, 2020.
>
> [8]: "Deep Neural Networks and Tabular Data: A Survey." IEEE transactions on neural networks and learning systems, 2022.

---

> > ### Comment · Reviewer_fzEk · 2024-11-24
> > **Raising Score Following Rebuttal**
> >
> > I appreciate the authors' detailed clarifications. They have thoroughly addressed most of my concerns, and based on their additional explanations and efforts during the rebuttal, I have decided to raise my score from 3 to 5.

---

> > > ### Author Response · Authors · 2024-11-25
> > >
> > > Thank you for your feedback and engagement with our work. We are glad to see that you have raised your score for our submission. We agree that a theoretical foundation is valuable for advancing the field of data distillation. It's worth noting that data distillation has predominantly been an empirically studied domain, with many existing works focusing on practical applications. While we recognize the importance of theoretical groundwork, developing a comprehensive theoretical foundation would require extensive additional research that extends beyond the scope of our current study. We believe this presents an exciting opportunity for future work in the field.
> > >
> > > During this rebuttal phase, we have:
> > >
> > > - Clarified the motivation/need for investigation of tabular data distillation and the terms used.
> > > - Added 5 baselines (4 deepcore + DATM) to add to the comparisons.
> > > - Provided a rough scalability analysis of TDColER and some directions on which the technique can be formalized.
> > >
> > > We would like to ask if there are any outstanding issues with our submission that are limiting you from considering our paper as acceptable. We appreciate any feedback you can provide to help us improve our submission further for you to find it acceptable, and thank you for your time.

---

### Official Review · Reviewer_6ZKe · 2024-11-01

**Soundness:** 2
**Presentation:** 4
**Contribution:** 3
**Rating:** 6
**Confidence:** 3

**Summary:**

This paper focuses on the dataset distillation for tabular data and proposes a tabular data distillation framework TDColER via column embeddings-based representation learning.  Furthermore, a tabular data distillation benchmark, TDBench, is constructed to evaluate this framework.

**Strengths:**

1. Good Novelty: The research topic is interesting. It is essential to investigate the dataset distillation technique for real-world applications.

2. Comprehensive Evaluation: The authors conduct a comprehensive evaluation of the framework on a diverse set of 23 datasets from OpenML with many schemes and classifiers.

3. Clear Presentation: The paper is well-organized and presents the proposed method and experimental results in a clear and concise manner.

**Weaknesses:**

1. While the paper mentions related work in the field of dataset distillation, it does not provide a detailed comparison with existing methods specifically designed for tabular data. I wonder whether the distilled dataset can improve the state-of-the-art methods.

2. The paper briefly mentions the use of column embeddings but does not provide a detailed explanation of how these embeddings are learned or their importance in the overall distillation process. This limits the understanding of the proposed method and its underlying mechanisms.

3. My biggest concern lies in the real-world task scenarios where models are trained solely on tabular data. As I understand, current research focuses on table-image multimodal data, utilizing tabular information to enhance models' image classification capabilities. The practical application of using tabular data alone seems unsuitable for dataset distillation, as tabular data are generally sensitive. Furthermore, tabular data occupy minimal storage space, raising the question of whether it is necessary to investigate dataset distillation for tabular data.

**Questions:**

Please refer to Weakness 3.

---

> ### Author Response · Authors · 2024-11-21
>
> We thank you for your detailed review and insightful comments. We have addressed each of your concerns pointed out in the weakness below. It would be great to hear your feedback on our responses.
>
> **Weakness 1: No comparison to existing methods designed for tabular data**
>
> Thank you for bringing up this point. This was in fact one of the main motivations behind this work. We are not aware of any established works that tackle dataset distillation/condensation and proposed a novel method specifically for tabular data. In fact, our goal is to establish such a baseline in tabular data, similarly to [1] in computer vision, for future researchers to propose and test potential state-of-art methods using our benchmarking suite.
>
> **Weakness 2: Missing details of column embeddings**
>
> Thank you for pointing this out. Column embeddings are indeed a crucial part of our proposed TDColER, and was adapted from the feature tokenzier method seen in [2]. We further discuss the details behind the training objective, fine-tuning process and the architecture of the encoders in Appendix A.4, which also discuss the column embedding module in more detail. We also include visual descriptions of the architectures in Figures 8,9,10,11 of the Appendix. In particular, Figures 8 and 9 describe how we handle the column embeddings explicitly. We will also edit the main manuscript to make this detail easier to find.
>
> **Weakness 3: Is distillation of tabular data alone worth it?**
>
> Thank you for raising this question. This is indeed a critical point to address when examining data distillation *solely* from the lens of uni-modal tabular data. We believe that the merit of dataset distillation is not limited to storage space. For example, our HPO use case experiment shows that conducting HPO with the distilled data on averages uses only 21.84\% of the runtime compared to using the full data, while yielding up to 98.37\% of the performance. In time-sensitive scenarios where some trade-off between computation time and performance is acceptable, using the distilled data can make a critical difference. In addition, there are also cases such as those involving privacy or copyright infringement, where the original data cannot be stored indefinitely, and eventually need to be discarded. In such cases, distilled data can also prove to be a valuable proxy that can provide a compromise between regulation/privacy and utility (e.g. downstream performance). We would also like to point to the fact that while many publicly available tabular datasets are *smaller* in storage compared to other modalities, there are still cases of large tabular data that exists in internal resources of many organizations. One good example is web data, where there is a constant stream of user data every day. We argue that one of the main roadblocks that prevent a full utilization of such large data is the scale, which dataset distillation can help with.
>
> **References**
>
> [1]: "DC-BENCH: Dataset Condensation Benchmark." NeurIPS 2022.
>
> [2]: "Revisiting Deep Learning Models for Tabular Data." NeurIPS 2021.

---

> > ### Comment · Reviewer_6ZKe · 2024-11-23
> >
> > Thank you for the author's response. However, there remains doubt whether the benchmark alone is suitable for ICLR, given that tabular dataset distillation is a relatively niche topic. It seems the author has not revised the manuscript; ideally, the rebuttal for ICLR should include revisions to demonstrate improvements in the paper's quality. Meanwhile, I remain doubtful about the privacy protection claim related to tabular data distillation. It would be best to provide a case study demonstrating the extent of such privacy protection, as balancing privacy and knowledge in tabular data poses a challenging hurdle.
> >
> > Although other reviewers are not optimistic about this paper, and I have taken their suggestions into consideration, I have decided to keep my score due to the open-source benchmark contribution. However, I reserve my recommendations regarding its novelty, significance, and experiments.

---

> > > ### Author Response · Authors · 2024-11-24
> > >
> > > Thank you for your feedback and suggestions. We would like to address a few key points you raised:
> > >
> > > **Suitability for ICLR**: You make a fair point that tabular dataset distillation may be seen as a niche topic. However, we would like to first note that _it is a niche topic because it hasn't been explored before_. In addition, another main contribution we claim is the **usage of learned presentations** that significantly improve downstream performance -- and thus distillation quality. In that sense, we feel that this work can be of value to ICLR.
> > >
> > > **Manuscript revisions**: We have actually updated the manuscript with requested changes -- we added clarifications and results from additional experiments at the end of the appendix for ease of viewing.
> > >
> > > **Privacy protection claims** : We agree this is an area that warrants further investigation and evidence. Your suggestion of including a case study demonstrating the privacy protection aspects is excellent. As you have rightly noted, balancing the information and privacy in tabular data can be quite challenging, which is also why we believe that this topic deserves an in-depth analysis and was a bit out of scope for this work. In addition, it can become a quite complex question to answer fully due to the number of steps and components involved in the distillation process. In this work, we did not conduct such analysis, but we feel that this is a very promising future work.
> > >
> > > **Novelty**: We do not necessarily propose a "new" distillation scheme, but we show how the  application of distillation scheme naively to tabular data does not work, and our novelty is in setting up a tabular representation pipeline that is able to learn high fidelity representations, that boost the performance of almost all existing distillation schemes on tabular data (24.75-105.80% for $k$-means, 19.92-108.79% for agglomerative, 16.82-82.32% for GM and 5.72-44.96% for KIP at IPC=10). Our results also highlight this novel insight -- that distillation schemes that perform well with vision data and models do not perform as well with tabular data and models (even with the representation learning) when compared to simplistic distillation schemes such as $k$-means clustering (median regret of 0.4056 for $k$-means vs 0.9415 of KIP or 0.7952 of GM as seen in Table 3).
> > >
> > > **Significance**: We believe that we provide significant improvements for tabular data distillation tasks, improving performance of models trained on the smallest distillation size by 15.82-108.79% across the board (from 15.82-36.00% for gradient boosted trees to 44.96-108.79% for nearest neighbor models). We think these are significant improvements for tabular data, enabled by our novel tabular representation learning pipeline for distillation. Furthermore, we demonstrate how this distillation can benefit downstream tasks such as HPO, allowing us to get to 98.37% of the best possible performance at 21.84% the computational cost.
> > >
> > > **Experiments**: We believe that we have thoroughly evaluated our proposed scheme over our proposed benchmark, covering 23 datasets, 6 tabular encoders, 4 distillation schemes, 7 downstream tabular models, resulting in 226k distillation datasets, and 540k+ downstream model training. This is a more elaborate evaluation than any considered in existing distillation (including the DC bench benchmarking paper). In addition, we have also evaluated 5 new distillation schemes suggested by other reviewers in this discussion phase.
> > >
> > > Thank you again for your constructive feedback. We appreciate your recognition of the our contributions and suggestions for improvement!

---

### Official Review · Reviewer_vNFz · 2024-11-03

**Soundness:** 2
**Presentation:** 3
**Contribution:** 2
**Rating:** 5
**Confidence:** 4

**Summary:**

This paper studies the problem of tubular data distillation, and provides a comprehensive benchmark for existing models and datasets.

**Strengths:**

1. Tabular dataset distillation is an important and interesting problem.
2. The paper evaluates a large number of datasets, methods and different model architectures, and provides valuable insights.
3. The paper is well-written and easy to follow.

**Weaknesses:**

1. Limited Novelty.  While the paper addresses an important issue, the proposed framework includes column embedding based representation learning and a distillation function, both appear standard and pre-existing.  Combing them together appears a bit straightforward as well.   In addition, [1] also proposes an algorithm which project dataset into a latent space for data distillation. It is suggested that the paper discuss in-depth the novelty by comparing with recent works such as [1].

2. In Section 2, the paper points out two main characteristics for tabular data, but it is unclear how the proposed framework addresses these challenges.

3. All the distillation methods considered are proposed before 2022. The paper also only compares its framework with the off-the-shelf distillation method. The authors should add more recent distillation methods (2022-2024), and compare with more recent baselines, such as [1] and baselines in Table 1 of [1] .

Given the above concerns,  the authors may consider re-position the paper as a benchmark paper.

[1] Effective Data Distillation for Tabular Datasets, AAAI-24

**Questions:**

how the proposed framework addresses the challenges unique for tabular data, exactly? That is, which part of the design is specifically applicable for tabular data?

---

> ### Author Response · Authors · 2024-11-21
>
> We thank you for your detailed review and insightful comments. We have addressed each of your concerns pointed out in the weakness below. It would be great to hear your feedback on our responses.
>
> **Weakness 1: Limited novelty, difference from previous work**
>
> Thank you for pointing this out. In addition to all the methods and pipelines covered in [1], this work considers an additional transformer-based autoencoder architecture and and the Gradient Matching (GM) distillation method. In particular, while our finding also shows that using the latent space is useful, we further back this finding empirically by conducting additional ablation studies on affects of different binning schemes. It is also worth noting that [1] is a two-page student abstract and has no code artifacts, while we provide a much more in-depth analysis and a complete python package that can be used by the community. We have also added a detailed comparison of the difference between this work and [1] which can be found in Appendix D.3.
>
> **Weakness 2: Unclear how the proposed framework addresses challenges for tabular data**
>
> Our proposed method, TDColER, aims to address feature heterogeneity by employing a uniform binning strategy to process the input data. TDColER ensures that the model can effectively learn from the data, regardless of the feature type. The column embeddings of TDColER allows for native handling of numerical and categorical data, as well as missing data. We also show empirically that using the column embeddings greatly improves the downstream performance of all families of distillation methods -- clustering, gradient-based and kernel-based -- considered in benchmark. Additionally, the model agnosticity challenge is addressed by leaving a flexible choice of distillation algorithm. In addition, depending on the type of models, we also leave the option to use either the dense (embedding) representation or the original representation for downstream tasks. This flexibility allows for a fair comparison between differentiable and non-differentiable models, and allows users to choose the most suitable representation based on the specific requirements of their application. Our experiments show that while the usage of latent embeddings for downstream classification is shown to be helpful in most cases, model-specific distillation methods borrowed from computer vision are consistently outperformed by simpler clustering-based baselines, again highlighting the differences we note between tabular and vision data. Our results justify the need for further development of model-independent distillation schemes that are targeted towards tabular data.

---

> ### Author Response · Authors · 2024-11-21
>
> **Weakness 3: More recent distillation methods**
>
> We apologize for any confusion in the naming convention. Our results contain all results seen in [1], including that of Table 1. However, we have modified how we refer to some of the methods (e.g. NNC $\to$ KNearestNeighbors) that may have lead to some confusion. Additionally, although we have results for Naive Bayes Classifier (NBC), we drop it from the analysis because the classifier shows inferior performance on full data, which lead to an inflated performance on the distilled datasets when aggregating with other classifiers. This can be verified in the results csv files that are included in the supplementary materials.
>
> During this rebuttal phase, we also have added four representative deep-coreset-selection methods from [2] as well as a more up-to-date gradient-based distillation methods from [3] and [4] during this rebuttal period. We would also like to highlight that the TDBench framework package allows for quick prototyping of new distillation methods and can be easily extended. The detailed results from these additional comparisons can be found in the modified Appendix D.1.
>
> **Question: Which part of the design is specifically applicable for tabular data?**
>
> The design of TDColER is specifically tailored to address the challenges unique to tabular data by using of column embeddings to represent both categorical and numerical features. The column embedding-based representation learning allows the autoencoder to capture the relationships between different columns. By learning the latent projection of the dataset, TDColER enhances the effectiveness of the distillation algorithm, reducing the data size while preserving the essential information. This is particularly important for tabular data, where the number of features can be large, and the data size can be substantial. In addition, our analysis shows that the conversion of the dataset from a sparse one-hot representation to a dense space is beneficial for both distance-based distillation algorithm (clustering) and NN-based algorithms (GM, KIP, MTT, DATM, deepcore).
>
> Another key aspect we consider is that the best performing models for tabular data are usually not NN-based models [5,6]. Another key contribution in our work is the ablation tests to check which distillation methods are most applicable to such models. Our finding shows that while the learned latent space is useful, NN architecture-specific distillation methods developed for vision data are not so useful for SotA models like XGBoost, which we believe is a valuable contribution to the community.
>
> **References:**
>
> [1]: "Effective Data Distillation for Tabular Datasets." AAAI, 2024
>
> [2]: "Deepcore: A comprehensive library for coreset selection in deep learning." International Conference on Database and Expert Systems Applications, 2022.
>
> [3]: "Dataset Distillation by Matching Training Trajectories." CVPR, 2022.
>
> [4]: "Towards Lossless Dataset Distillation via Difficulty-Aligned Trajectory Matching." ICLR 2024.
>
> [5]: "Why do tree-based models still outperform deep learning on typical tabular data?" NeurIPS Datasets and Benchmark Track, 2022.
>
> [6]: "When Do Neural Nets Outperform Boosted Trees on Tabular Data?" NeurIPS 2023.

---

> > ### Author Response · Authors · 2024-11-25
> >
> > Dear reviewer vNFz, during this rebuttal period, we have:
> >
> > - Addressed the difference of our work compared to `Effective Data Distillation for Tabular Datasets (Kang et al.)`
> > - Clarified how our proposed approach addresses the issue of feature heterogenity and model agnosticity for distilling tabular data.
> > - Added comparisons with more recent baselines (4 deepcore + DATM) to further demonstrate the effectiveness of our approach.
> >
> > We would love to hear your feedback on our updates and look forward to discussing any remaining concerns you may have. Thank you for your time and consideration.

---

> > ### Comment · Reviewer_vNFz · 2024-11-27
> >
> > I have read the authors' responses and appreciate the added evaluations and discussions. I raise my score accordingly.

---

> > > ### Author Response · Authors · 2024-11-28
> > >
> > > Thank you your feedback. We are glad that you have raised the score for our submission. We would like to ask if there are any further issues with our submission that are limiting you from considering our paper as acceptable. We appreciate any feedback you can provide to help us improve our submission, and thank you for your time.

---

### Official Review · Reviewer_em7K · 2024-11-11

**Soundness:** 3
**Presentation:** 3
**Contribution:** 3
**Rating:** 6
**Confidence:** 3

**Summary:**

This work proposes a data distillation approach dedicated to tabular data. It uses the column representation to guide tabular data distillation. Specifically, this work first trains an encode to transform the original data to an intermediate representation for each column, then trains a decoder to recover the original data based on the intermediate representations. The distillation goal is to minimize the difference of intermediate column representation between the original and distilled tabular data. The comprehensive empirical results demonstrate the effectiveness of the proposed approaches.

**Strengths:**

1. This work investigate a important but less explored data modality, i.e., tabular data, in AI applications.
2. This work conduct a comprehensive assessment on 226,200 distilled dataset and 541,980 models trained on it.
3. The paper is easy to follow.

**Weaknesses:**

1. Although this paper conducts numerous experiments, the evaluation is based on a custom metric, relative regret, rather than on common metrics such as accuracy or mean squared error.

2. I think it is valuable to consider whether the distillation process impacts inter-column correlations, which are crucial for downstream tasks in tabular data, such as data analysis. Specifically, an ablation study could investigate if the distillation process preserves or reduces the correlation between columns, potentially affecting the dataset’s utility for the downstream applications.

3. It would strengthen this work if the authors could include a security and privacy analysis of their proposed TDColER. I am also curious whether the distilled dataset might introduce biases or vulnerabilities to adversarial attacks or privacy leakage.

4. The baseline comparison does not seem comprehensive. A comparison with other data distillation-like techniques, such as coreset selection [1], is missing.



[1] Guo, Chengcheng, Bo Zhao, and Yanbing Bai. "Deepcore: A comprehensive library for coreset selection in deep learning." International Conference on Database and Expert Systems Applications. Cham: Springer International Publishing, 2022.

**Questions:**

see weakness

---

> ### Author Response · Authors · 2024-11-21
>
> We thank you for your detailed review and insightful comments. We have addressed each of your concerns pointed out in the weakness below. We would love to hear your feedback on our responses.
>
> **Weakness 1: The evaluation is based on a custom metric, relative regret, rather than on common metrics such as accuracy or mean squared error.**
>
> Thank you for pointing this out. We agree that using regret as the primary evaluation metric may not be intuitive for all readers. However, we chose this metric because it allows us to make a fair comparison between the performance of different distillation methods across multiple datasets with respect to random sampling. We initially conducted our analysis by averaging the raw balanced accuracy score, but found that the results can be misleading in certain cases. This approach is inspired by [2], which benchmarks AutoML packages and aggregates the per-dataset performance scaled by the performance of random forest model to model the *performance gain* that occurs from different packages. Our regret metric is defined as the following: $(A_F-A)/(A_F-A_{R10})$, where $A_F$ is the performance from full-data and $A_{R10}$ is the performance on 10 randomly sampled points. The normalizing term $A_F-A_{R10}$ is used to scale the difference between the full data and random sampling performance. Similar to [2], the intuition is as the following: since random sampling is the simplest form of distillation, we should be aware of how well it performs in that particular setting when looking at other distillation methods. This prevents cases where a distillation method's downstream performance appears artificially high.  In our analysis, we found that simple averaging over all of our 23 datasets can lead to misleading results. We will make this more clear in our footnote when introducing this metric.
>
> For example, consider the following case:
>
> - On dataset A, the full-data performance is 0.9, random sampling performance is 0.5, and method X's performance is 0.6.
> - On dataset B, the full-data performance is 0.7, random sampling performance is 0.5, and method Y's performance is 0.6.
>
> At a glance, it may appear that method X and method Y yield the same performance. However, this is not the case in reality, as dataset B is a much *harder* dataset. By computing the regret instead, we are able to get a fairer understanding of how each methods perform with respect to the lower (random sample@IPC=10) and upper (full data) bounds.
>
>
> **Weakness 2: Analysis of impact on inter-column correlations**
>
> We appreciate you bringing up this angle, and we agree that this is a very interesting topic to further branch into. In this work, we mainly focus on downstream performance based on the assumption that if there are correlations between features that are *useful* for the downstream task, losing such correlations will be reflected in degraded downstream performance. In addition, we also report very high reconstruction accuracy from our autoencoders, which also suggests that the correlations are indeed picked up by the encoder and decoders. Another complication is that the best downstream performance occurs when the distilled data is in the *latent* space, which makes it difficult to conduct a 1-on-1 comparison against features in the original space. This said, we do agree that an in-depth analysis to confirm whether this indeed happens is valuable. We conducted some preliminary analysis into whether our best performing pipelines preserve such pairwise feature correlations and have added it to Appendix D.2. Please refer to the modified text for further details into this analysis.

---

> ### Author Response · Authors · 2024-11-21
>
> **Weakness 3: Security, privacy and bias analysis of TDColER**
>
> We thank you for raising this valuable point. This is indeed an interesting direction for data distillation and a very natural extension of this work. However, we believe that the scope of this question can be quite broad as it can be examined from many possible use cases and components to be examined (leakage from embeddings, distillation method, decoding step, etc.) and that it deserves a dedicated work to be fully answered. This work aims to serve as a starting point for data distillation in the tabular data domain. Following suite from previous work that examined data distillation methods [3], we focus on comparing the downstream performance of the distilled data. Extending upon this, future works can examine best performing methods identified from our framework from the lens of privacy/bias mitigation against downstream utility. Simple analysis includes examining robustness against attacks like membership inference (privacy), or examining and mitigating bias on sensitive attributes by modifying distillation algorithms to use weighted sampling methods. These are just some ideas we could brainstorm, but we believe that there is much more to these questions that can reveal interesting insights. Again, thank you for raising this question, and we hope to answer it in our following works.
>
> **Weakness 4: Lack of comparison with existing data condensation methods (Deepcore)**
>
> Thank you for this suggestion. We have incorporated 4 representative methods  from [1] (forgetting, GraNd, Glister and Graph Cut) and 2 additional gradient trajectory-matching methods [5,6] to our benchmarks and show the results in Appendix D.1. Due to limited time during the author-reviewer discussion, we only consider XGBoost classifier with TF-SFT encoder. We find that the results among the four methods on our framework is consistent to the results reported in [1] -- Graph Cut shows best performance overall while other methods are indistinguishable over different datasets. However, we also observe that the Deepcore methods, which are based on specific NN architectures, do not generalize well for non-differentiable models like XGBoost and also fall behind the gradient-matching methods from [4,5,6].
>
> We would also like to bring to attention that one of our contributions -- the TDBench framework -- was designed with extensibility in mind and will be open to public and anyone can add their own methods to compare against our results (as we were able to incorporate these 4 deepcore distillation schemes and 2 trajectory-matching schemes).
>
> **References:**
>
> [1]: "Deepcore: A comprehensive library for coreset selection in deep learning." International Conference on Database and Expert Systems Applications, 2022.
>
> [2]: "Amlb: an automl benchmark." Journal of Machine Learning Research, 2024.
>
> [3]: "DC-BENCH: Dataset Condensation Benchmark." NeurIPS Datasets and Benchmark Track, 2022.
>
> [4]: "Dataset Condensation with Gradient Matching.", ICLR, 2021.
>
> [5]: "Dataset Distillation by Matching Training Trajectories." CVPR 2022.
>
> [6]: "Towards Lossless Dataset Distillation via Difficulty-Aligned Trajectory Matching." ICLR 2024.

---

> > ### Comment · Reviewer_em7K · 2024-11-25
> >
> > Thank you for your response. Although some of my concerns have been addressed, I am still not convinced by the explanation of the comparison metrics and baseline experiements. I think the authors should include additional comparisons using common metrics to more fairly demonstrate the effectiveness of their approaches. However, I will adjust my score to credit the authors' new experiments on feature correlations.

---

> ### Author Response · Authors · 2024-11-25
>
> Thank you for your feedback and engagement with our work. We appreciate your recognition of the new experiments we have conducted on feature correlations. We agree that reporting the raw values can provide a more intuitive way to understand the results. We have updated the appendix to include additional comparisons using our raw metric (balanced accuracy score) to the appendix to further back demonstrate the effectiveness of 1. using TDColER to distill, 2. distilling with simpler clustering baselines compared to sophisticated vision-based methods. Tables 19~24 of Appendix D.4 show the raw values and each method's win rates vs. random sampling and the number of instances each method ranks best for the dataset/classifier setting. Please note that only table 19 includes the methods that were added during the rebuttal period due to time constraints. We will run the rest of the experiments and add them in the future. These tables show that applying TDColER instead of naive distillation indeed raises the win-rate of all distillation methods against random sampling. It also shows that agglomerative and $k$-means clustering show superior performance to vision-based distillation methods. These results are consistent with our aggregate results shown using the regret score in the main manuscript. We hope that these additions address your concerns and improve the overall quality of our submission. We appreciate your time and feedback, and thank you for adjusting your score. We would appreciate it if you could share whether you feel there are other outstanding issues with our submissions that can be amended.

---

### Author Response · Authors · 2024-11-26

We would like thank all the reviewers for their helpful feedback and suggestions.

### Summary

We are glad to see that reviewers appreciate the following aspects of our work:

- Exploring data distillation for tabular data [em7K,vNFz,6ZKe], a less explored domain [em7K]
- Comprehensiveness of our experiments and analysis [em7K,vNFz,6ZKe,fzEk].
- Our contribution in the form of a open-source benchmarking package [6ZKe,fzEk].

### Updates

During this rebuttal period, we have:

- Conducted additional analysis on preservation of inter-column correlations [em7K]
- Added 6 baselines (4 from Deepcore, DATM and MTT) -- [em7K,vNFz,fzEk]
- Provided a rough scalability analysis of TDColER and some directions on which the technique can be formalized [fzEk]
- Added a comparison that uses raw metric scores instead of aggregation [em6K]

All the updates have been reflected (highlighted in blue) in the appendix of the manuscript.

---

### Author Response · Authors · 2024-12-02

Dear Reviewers and Area Chair,

Thank you very much for taking the time to read our revised manuscript! We appreciate the continued engagement and the valuable feedback/comments we received during this period. We are very happy to see that reviewers **em7K**, **vNFz** and **fzEk** are satisfied with our responses and have raised their scores during this discussion phase.

We are thankful to the reviewers for raising their scores. If the reviewers are satisfied with our updates and responses, and believe that our work is worthy of acceptance, we kindly ask them to consider rating our paper as a "full accept". We remain available to resolve any outstanding issues that would improve our paper from a borderline to an accept in the remaining time allotted for discussion.

Thank you again for your time and consideration,
Authors

---

### Meta-Review · Area_Chair_UN2N · 2024-12-22

**Metareview:**

The paper introduces TDColER, a novel framework designed for distilling tabular data. It addresses the challenges of feature heterogeneity and non-differentiable models commonly used with tabular data. The authors propose using column embeddings and representation learning to enhance the quality of distilled data. The paper also introduces TDBench, a benchmark for evaluating distillation methods on tabular data. Through extensive experiments, the authors demonstrate TDColER's ability to significantly improve the quality of distilled datasets across various models and datasets.


The reviewers are satisfied with the responses by the authors. However, their enthusiasm toward this paper is unfortunately rather lukewarm. The reasons could be the benchmarks alone might not be novel enough for the high bar of ICLR. Also, some of the reviewers remain doubtful about the privacy protection claim related to tabular data distillation. It would be best to provide a case study demonstrating the extent of such privacy protection, as balancing privacy and knowledge in tabular data poses a challenging hurdle.

**Additional Comments On Reviewer Discussion:**

There were active discussions between authors and reviewers. however, the reviewers' new score seem to indicate that their enthusiasm toward this paper is rather lukewarm, possibly due to the lack of novelty of this work.

---

### Decision · Program_Chairs · 2025-01-22

Reject